# Optimal Transport for Time-Series Imputation

**Hao Wang**[1]  **Zhengnan Li**[2]  **Haoxuan Li**[3]  **Xu Chen**[4]  **Mingming Gong**[5,6]
**Bin Chen**[7]  **Zhichao Chen**[1,*]

[1]Zhejiang University  [2]Communication University of China  [3]Peking University
[4]Renmin University of China  [5]The University of Melbourne
[6]Mohamed bin Zayed University of Artificial Intelligence
[7]University of Chinese Academy of Sciences

Ho-ward@outlook.com  fmlyd@cuc.edu.cn  justuschencse@zju.edu.cn

## Abstract

Missing data imputation through distribution alignment has demonstrated advantages for non-temporal datasets but exhibits suboptimal performance in time-series applications. The primary obstacle is crafting a discrepancy measure that simultaneously (1) *captures temporal patterns*—accounting for periodicity and temporal dependencies inherent in time-series—and (2) *accommodates non-stationarity*, ensuring robustness amidst multiple coexisting temporal patterns. In response to these challenges, we introduce the Proximal Spectrum Wasserstein (PSW) discrepancy, a novel discrepancy tailored for comparing two *sets* of time-series based on optimal transport. It incorporates a pairwise spectral distance to encapsulate temporal patterns, and a selective matching regularization to accommodate non-stationarity. Subsequently, we develop the PSW for Imputation (PSW-I) framework, which iteratively refines imputation results by minimizing the PSW discrepancy. Extensive experiments demonstrate that PSW-I effectively accommodates temporal patterns and non-stationarity, outperforming prevailing time-series imputation methods. Code is available at https://github.com/FMLYD/PSW-I.

## 1 Introduction

The incompleteness of time-series data is a widespread issue in many fundamental fields. For example, in healthcare (Prosperi et al., 2020), users utilize monitoring devices to track their health data, but these records may be incomplete due to occasional device failures or disconnections. Similarly, in manufacturing and process engineering (Wang et al., 2024a), automation equipment often includes sensors that monitor operational statistics for safety and efficiency, but these records can also be affected by human errors or sensor malfunctions. Such incompleteness damages data integrity, which is critical for accurate and reliable analytics (Qiu et al., 2024; 2025; Liu et al., 2024), highlighting the importance of effective Time-Series Imputation (TSI) techniques.

Deep learning methods have attracted substantial attention in TSI due to their ability to model nonlinearities and temporal dependencies. These methods can be classified into two categories: *predictive methods*, which estimate deterministic values for the missing entries within the time-series (Cao et al., 2018; Du et al., 2023), and *generative methods*, which generate missing values conditionally given observed ones (Luo et al., 2018). Despite their effectiveness, they are primarily challenged by model selection amidst incomplete data (Jarrett et al., 2022), and require masking some observed entries during training to generate labels (Chen et al., 2024), which can limit performance especially given high missing ratios.

To counteract the defects with deep imputation methods, alignment-based methods have emerged as an alternative. These methods eliminate the need for masking observed entries and training parametric models on incomplete data, offering advantages in sample efficiency and implementation simplicity. While alignment-based methods have proven to be effective for imputing missing data in non-temporal datasets (Muzellec et al., 2020; Zhao et al., 2023; Wang et al., 2024a;b), their application to TSI is challenging and remains largely unexplored. Our experiments (see Section 4.2) indicate that directly

---

*Corresponding author.

applying existing alignment-based methods to temporal data yields poor performance. Therefore, how to adapt distribution alignment methods to TSI remains a challenging and open problem.

In distribution alignment, the choice of discrepancy measure matters, which should accommodate the dataset characteristics (Courty et al., 2017; Wang et al., 2023; Liu et al., 2022b). Typically, time-series are uniquely characterized by temporal patterns, such as periodicities and temporal dependencies, and often exhibit non-stationary fluctuations. Motivated by this, the key to accommodate the alignment-based imputation methods to TSI is devising a discrepancy measure that captures temporal patterns while accommodating non-stationarity in time-series.

To this end, we propose the *Proximal Spectrum Wasserstein (PSW)* discrepancy, a discrepancy tailored to compare *sets* of time-series based on optimal transport. Specifically, PSW integrates a Pairwise Spectral Distance (PSD), which transforms time-series into the frequency domain and then calculates the pair-wise absolute difference. By comparing time-series in the frequency domain, the underlying temporal dependencies and patterns are captured. Moreover, PSW incorporates Selective Matching Regularization (SMR), which relaxes the hard matching constraints of traditional optimal transport to enable flexible mass matching between distributions. This relaxation enhances robustness to non-stationarity. Building upon PSW, we propose the PSW for Imputation (PSW-I) framework, which iteratively refines the imputed missing values by minimizing the PSW discrepancy. Extensive experiments demonstrate that PSW-I effectively captures temporal patterns and accommodates non-stationarity and significantly outperforms existing time-series imputation methods.

**Contributions.** The key contributions of this study are summarized as follows:

- We propose the PSW discrepancy, which innovatively extends optimal transport to compare distributions of time-series by encapsulating temporal patterns and accommodating non-stationarity.

- We develop PSW-I, the first alignment-based method for TSI. It eliminates the need for masking observed entries during training and the complexities of training parametric models on incomplete data, enhancing sample efficiency and ease of operation.

- We conduct comprehensive experiments on publicly available real-world datasets to validate the effectiveness of PSW-I, demonstrating its superiority over existing TSI methods.

## 2 PRELIMINARIES

### 2.1 PROBLEM FORMULATION

Suppose $\mathbf{X}^{(\mathrm{id})} \in \mathbb{R}^{\mathrm{N} \times \mathrm{D}}$ is the ideally complete time-series with N chronologically-ordered observations and D features. The missing entries are indicated by a binary matrix $\mathbf{M} \in \{0, 1\}^{\mathrm{N} \times \mathrm{D}}$, where $\mathbf{M}_{n,d}$ is set to 1 if the corresponding entry $\mathbf{X}_{n,d}^{(\mathrm{id})}$ is missing, and 0 otherwise. Consequently, the observed dataset $\mathbf{X}^{(\mathrm{obs})}$ can be obtained via the Hadamard product: $\mathbf{X}^{(\mathrm{obs})} := \mathbf{X}^{(\mathrm{id})} \odot (1 - \mathbf{M}) + \mathrm{nan} \odot \mathbf{M}$. The goal of TSI is to construct an imputed data matrix $\mathbf{X}^{(\mathrm{imp})} \in \mathbb{R}^{\mathrm{N} \times \mathrm{D}}$ based on the observed entries in $\mathbf{X}^{(\mathrm{obs})}$, such that $\mathbf{X}^{(\mathrm{imp})} \approx \mathbf{X}^{(\mathrm{id})}$.

### 2.2 OPTIMAL TRANSPORT

Optimal Transport (OT) is a mathematical tool that quantifies the discrepancy between two probability distributions by finding the least-cost plan to transform one distribution into the other. Originally proposed by Monge (1781), the formulation involved finding an optimal mapping between two continuous distributions. However, this original formulation posed challenges related to the existence and uniqueness of solutions. Addressing these issues, Kantorovich (2006) proposed a more computationally feasible formulation in Definition 1, a convex programming problem solvable via the simplex algorithm (Disser and Skutella, 2019).

**Definition 2.1.** *Consider empirical distributions $\alpha = \alpha_{1:\mathrm{n}}$ and $\beta = \beta_{1:\mathrm{m}}$, each with n and m samples, respectively; we seek a feasible plan $\mathbf{T} \in \mathbb{R}_{+}^{\mathrm{n} \times \mathrm{m}}$ to transport $\alpha$ to $\beta$ at the minimum possible cost:*

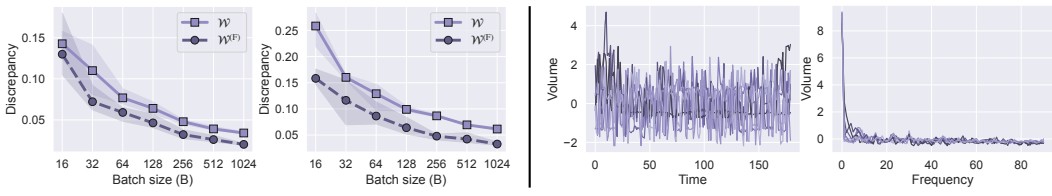

(a) Discrepancy on ETTh1 and Electricity.    (b) 8 temporal patches on ETTh1 and their spectra.

Figure 1: Case study on the discrepancies calculated in the time and frequency domains.

$$\mathcal{W}(\alpha, \beta) := \min_{\mathbf{T} \in \Pi(\alpha, \beta)} \langle \mathbf{D}, \mathbf{T} \rangle,$$

$$\Pi(\alpha, \beta) := \left\{ \begin{array}{l} \mathbf{T}_{i,1} + ... + \mathbf{T}_{i,\mathrm{m}} = \mathbf{a}_i, i = 1, ..., \mathrm{n}, \\ \mathbf{T}_{1,j} + ... + \mathbf{T}_{\mathrm{n},j} = \mathbf{b}_j, j = 1, ..., \mathrm{m}, \\ \mathbf{T}_{i,j} \geq 0, i = 1, ..., \mathrm{n}, j = 1, ..., \mathrm{m}, \end{array} \right. \tag{1}$$

*where $\mathcal{W}(\alpha, \beta)$ denotes the minimum transport cost, known as the standard Wasserstein discrepancy; $\mathbf{D} \in \mathbb{R}_+^{\mathrm{n} \times \mathrm{m}}$ represents the pairwise distances calculated as $\mathbf{D}_{i,j} = \|\alpha_i - \beta_j\|_2$; $\mathbf{a} = [\mathbf{a}_1, \ldots, \mathbf{a}_\mathrm{n}]$ and $\mathbf{b} = [\mathbf{b}_1, \ldots, \mathbf{b}_\mathrm{m}]$ are the masses of samples in $\alpha$ and $\beta$, respectively.*

## 3 METHODOLOGY

### 3.1 MOTIVATION

Distribution alignment has proven effective for imputing missing data in non-temporal datasets (Muzellec et al., 2020; Zhao et al., 2023), offering advantages in sample efficiency and implementation simplicity. These methods operate by iteratively sampling subsets of the incomplete dataset and updating missing entries to minimize distributional discrepancies between these subsets. This ensures that the imputed values maintain statistical properties consistent with the entire dataset, grounded in the assumption that different subsets from the same dataset share the same distribution.

However, applying distribution alignment to TSI poses significant challenges. Our experiments (see Section 4.2) indicate that existing alignment-based methods perform poorly on temporal data. In the field of domain adaptation (Courty et al., 2017; Liu et al., 2022b), it is well-recognized that the effectiveness of distribution alignment heavily depends on the choice of discrepancy measure, which must be tailored to the specific properties of the data and the task. Therefore, we aim to refine the discrepancy measure to accommodate the unique characteristics of time-series data, for enhancing TSI performance. Importantly, there are several questions that need to be answered. Do existing discrepancy measures accommodate the characteristics, such as the temporal patterns and non-stationarity, in time-series data? *How to design a discrepancy measure for comparing distributions of time-series?* Does designed discrepancy improve imputation performance?

### 3.2 PAIRWISE SPECTRUM DISTANCE FOR TEMPORAL PATTERN ENCAPSULATION

Time-series data are characterized by temporal patterns, which reflects correlations between different time steps and provide rich semantic information essential for comparing time-series. The canonical Wasserstein discrepancy ($\mathcal{W}$) fails to capture these temporal patterns because the pairwise distance is computed on a step-wise basis, treating the observations at each step independently and disregarding temporal correlations. A simple modification might involve using a patch-wise distance: employing a sliding window to generate temporal patches of size $\mathrm{T}$ and subsequently computing distances between patches. However, this method still treats different steps within the patch individually, failing to measure dissimilarity between patches in a way that encapsulates their temporal patterns.

To address this limitation, we propose the Pairwise Spectrum Distance (PSD), which uses the Discrete Fourier Transform (DFT) to convert time-domain data into the frequency domain. The DFT decomposes each temporal patch into its spectral components (Wu et al., 2025; Wang et al., 2025),

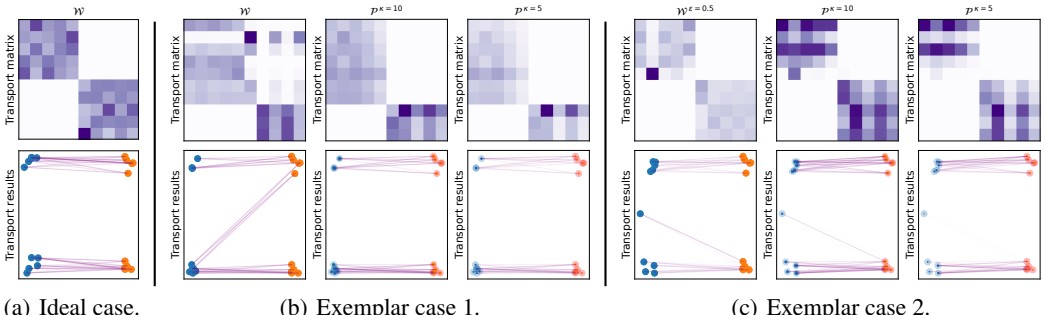

(a) Ideal case.  (b) Exemplar case 1.  (c) Exemplar case 2.

Figure 2: The impact of non-stationarity and the resilience of PSW to it. The toy dataset includes two batches of samples, each distinguished by color. Two modes co-exist due to non-stationarity, differentiated by the vertical positioning of the samples.

where each component corresponds to specific temporal patterns in the data. By comparing patches in the frequency domain, PSD effectively captures and compares the underlying temporal patterns. Building upon PSD, we propose the Spectrum-enhanced Wasserstein distance in Definition 3.1.

**Definition 3.1** (Spectrum-enhanced Wasserstein Distance)**.** The distance between two distributions $\alpha, \beta \in \mathbb{R}^{B \times T \times D}$ of temporal patches is defined as $\mathcal{W}^{(\mathrm{F})}(\alpha, \beta) = \min_{\mathbf{T} \in \Pi(\alpha, \beta)} \left\langle \mathbf{D}^{(\mathrm{F})}, \mathbf{T} \right\rangle$, where T is the patch size, $\mathbf{D}^{(\mathrm{F})}$ is the pairwise distance matrix with elements computed using the Pairwise Spectrum Distance: $\mathbf{D}_{i,j}^{(\mathrm{F})} = \|\mathcal{F}(\alpha_i) - \mathcal{F}(\beta_j)\|_1$, and $\mathcal{F}$ denotes the DFT operator.

**Case Study.** Fig. 1 (a) compares $\mathcal{W}$ and $\mathcal{W}^{(\mathrm{F})}$. Specifically, $\mathcal{W}^{(\mathrm{F})}$ consistently decreases with increasing batch sizes and achieves comparable performance of $\mathcal{W}$ with smaller batch sizes. For instance, the value of $\mathcal{W}^{(\mathrm{F})}$ with a batch size of 128 approximates the value of $\mathcal{W}$ with a batch size of 1024. This efficiency arises because spectrum captures the consistent patterns across patches, especially in periodic data where spectral amplitudes are stable despite significant changes in the time domain in Fig. 1(b). These findings demonstrate the advantage of $\mathcal{W}^{(\mathrm{F})}$ for time-series data, as it better captures temporal patterns.

### 3.3 SELECTIVE MATCHING REGULARIZATION FOR NON-STATIONARITY ROBUSTNESS

Time-series data often exhibit *non-stationarity*, characterized by time-varying patterns and sudden fluctuations. For instance, the Electricity dataset shows significant differences in consumption patterns between weekdays, weekends, and holidays; similarly, the Weather dataset displays distinct climatological patterns across different seasons. This non-stationarity produces multiple coexisting patterns or regimes within the data, complicating the accurate calculation of distributional discrepancies.

The canonical Wasserstein discrepancy (Definition 2.1) struggles in the presence of non-stationarity. As illustrated in Fig. 2, it may incorrectly pair patches from different modes, leading to inaccurate distributional discrepancies and misleading imputation updates. This issue arises due to the matching constraints that require matching of all masses of all samples (Wang et al., 2023). Consequently, assuming that a patch from a new mode, denoted as $\delta_z$, is added to $\alpha$, the matching constraints force a match between $\delta_z$ and patches in $\beta$, distorting the matching process and yielding an imprecise estimate of the discrepancy. This vulnerability is formalized in Lemma 3.2, which shows that the Wasserstein distance $\mathcal{W}$ increases as the added mode deviates further from the typical elements of $\beta$.

**Lemma 3.2.** *Suppose that $\tilde{\alpha} = \zeta \delta_z + (1 - \zeta)\alpha$ is a distribution perturbed by a Dirac mode at $z$ with relative mass $\zeta \in (0, 1)$. For a sample $y^*$ in the support of $\beta$, Fatras et al. (2021) demonstrate:*

$$\mathcal{W}(\tilde{\alpha}, \beta) \geq (1 - \zeta)\mathcal{W}(\alpha, \beta) + \zeta \left( D(z, y^*) - g(y^*) + \int g \, d\beta \right)$$

*where $D(z, y^*)$ is the deviation of $\delta_z$, $g$ is the optimal dual potential of $\mathcal{W}(\alpha, \beta)$.*

To enhance robustness to non-stationarity, it is plausible to relax the marginal matching constraints, allowing for matching a flexible mass of each patch. Inspired by weak transport principles (Chizat

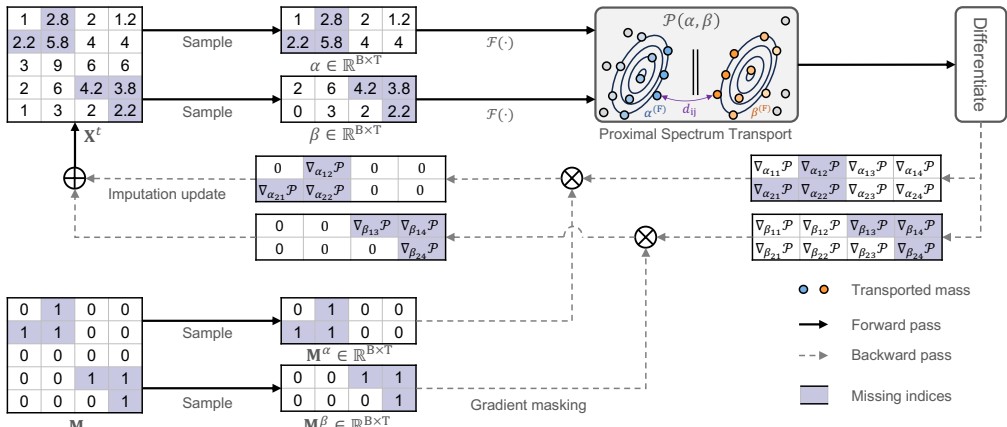

Figure 3: The workflow of PSW-I with $p_{\mathrm{miss}} = 0.3$. The batch size ($B$) is set to 2 and the patch size ($T$) is set to 4. The number of features ($D$) is omitted for clarity.

et al., 2018; Séjourné et al., 2019), we introduce the *Proximal Spectral Wasserstein (PSW)* discrepancy, which replaces the hard marginal constraints with Selective Matching Regularization (SMR). This approach removes the requirement to match all mass between distributions, thus accommodating the coexistence of multiple modes and enhancing robustness to non-stationarity.

**Definition 3.3** (Proximal Spectral Wasserstein Discrepancy). *The PSW discrepancy seeks a transport plan* $\mathbf{T} \in \mathbb{R}_+^{n \times m}$ *that transports the distribution* $\alpha$ *to* $\beta$ *at minimal cost, defined as:*

$$\mathcal{P}^\kappa(\alpha, \beta) := \min_{\mathbf{T} \geq 0} \left\langle \mathbf{D}^{(\mathrm{F})}, \mathbf{T} \right\rangle + \kappa \left( \mathrm{D}_{\mathrm{KL}}(\mathbf{T}\mathbf{1}_m \| \Delta_n) + \mathrm{D}_{\mathrm{KL}}(\mathbf{T}^\mathsf{T}\mathbf{1}_n \| \Delta_m) \right) \tag{2}$$

*where* $\mathbf{D}^{(\mathrm{F})}$ *is the pairwise distance matrix computed using PSD;* $\kappa$ *is the matching strength;* $\Delta_n = \mathbf{1}_n/n$ *and* $\Delta_m = \mathbf{1}_m/m$ *are uniform simplex vectors;* $\mathcal{P}^\kappa$ *denotes the PSW discrepancy.*

**Case Study.** To illustrate the limitations of the existing Wasserstein discrepancy and the robustness of PSW to non-stationarity, we conduct a case study shown in Fig. 2. Ideally, in the absence of non-stationarity, Wasserstein discrepancy based on OT accurately matches patches within the same mode and correctly estimate the distributional discrepancy. However, in the presence of non-stationarity, we investigate two exemplary cases:

- **Example 1:** When the proportions of different modes vary across batches, standard OT incorrectly matches patches from different modes due to its force to match all mass. This leads to a biased Wasserstein discrepancy. In contrast, $\mathcal{P}$ employs a selective matching strategy, focusing on typical patches and effectively avoiding false matches across modes given small matching strength $\kappa \leq 10$.
- **Example 2:** When a newly emerging mode exists in a non-overlapping area between two modes, standard OT forces matching of the outlier mode with others, leading to inappropriate pairings and interfering with the matching of other patches. Conversely, $\mathcal{P}$ resists the interference as $\kappa$ decreases. When $\kappa$ is reduced to 5, the outlier mode is effectively excluded from the matching process.

Our approach differs from the entropic unbalanced optimal transport (Séjourné et al., 2019; Fatras et al., 2021), which also relaxes matching constraints through regularization, by omitting entropic regularization from formulation. This omission is crucial, as entropic regularization has been shown to hinder missing value imputation (Chen et al., 2024). Notably, without entropic regularization, the OT associated with PSW cannot be solved using the Sinkhorn algorithm. Instead, we employ the majorization-minimization algorithm (Chapel et al., 2021) to solve the OT problem. In Section 4.4, we present a rigorous comparative study that highlights the advantages of PSW over traditional entropic unbalanced optimal transport methods in the context of TSI.

### 3.4 PSW-I: PSW FOR TIME-SERIES IMPUTATION

While PSW effectively compares and balances distributions of temporal patches, it does not directly perform time-series imputation. To fill this gap, we propose the PSW for Imputation (PSW-I)

framework, which iteratively minimizes the PSW discrepancy between batches of patches to refine the imputation of missing values. The core steps of PSW-I are outlined in Fig. 3 and explained below.

**Initialization.** The incomplete dataset $\mathbf{X}^{(\text{obs})}$ is initialized by filling each missing entry with the average of its nearest observed steps, producing an initial imputation matrix $\mathbf{X}^{t=0}$. The imputed values are treated as learnable parameters, and their gradients are tracked in subsequent steps.

**Forward Pass.** Two batches of temporal patches, denoted as $\alpha \in \mathbb{R}^{\text{B} \times \text{D} \times \text{T}}$ and $\beta \in \mathbb{R}^{\text{B} \times \text{D} \times \text{T}}$, are sampled from the current imputed dataset $\mathbf{X}^t$ with batch size B. The PSW discrepancy $\mathcal{P}$ is then computed between these two batches according to Algorithm 1.

**Backward Pass.** The gradients of the PSW discrepancy $\mathcal{P}$ with respect to $\alpha$ and $\beta$ are calculated as follows using automatic differentiation:

$$\frac{\partial \mathcal{P}}{\partial \alpha_i} := \sum_{j=1}^{\text{m}} \mathbf{T}_{i,j} \mathbf{W}^{(\text{F})} \cdot \text{sign}(\mathbf{e}_{i,j})^\top, \quad i = 1, 2, \ldots, \text{B},$$

$$\frac{\partial \mathcal{P}}{\partial \beta_j} := -\sum_{i=1}^{\text{n}} \mathbf{T}_{i,j} \mathbf{W}^{(\text{F})} \cdot \text{sign}(\mathbf{e}_{i,j})^\top, \quad j = 1, 2, \ldots, \text{B},$$

where $\mathbf{W}^{(\text{F})}$ denotes the DFT matrix[1], $\mathbf{e}_{i,j} = (\alpha_i - \beta_j)\mathbf{W}^{(\text{F})} \in \mathbb{R}^{\text{T}}$. These gradients are used to update the imputed values in $\alpha$ and $\beta$ via gradient descent with an update rate $\eta$. Only the imputed values are updated during this process. PSW-I iteratively executes the forward and backward passes until hitting the early-stopping criteria on the validation dataset.

**Theoretical Justification.** We demonstrate that the *PSW discrepancy is a valid discrepancy measure* and satisfies the properties of a metric under mild conditions (Theorem C.1). Furthermore, we prove that *PSW is robust to non-stationarity in data* (Theorem C.2). Due to the convexity of the PSW discrepancy with respect to inputs $\alpha$ and $\beta$, *the convergence of PSW-I is guaranteed with error bounds (Theorem C.3)*. Detailed proofs are provided in Appendix C.

## 4 EMPIRICAL INVESTIGATION

### 4.1 EXPERIMENTAL SETUP

- **Datasets:** Experiments are performed on public time-series datasets (Wu et al., 2021; Liu et al., 2024), including ETT, Electricity, Traffic, Weather, PEMS03, Illness, and Exchange. To simulate point-wise missingness (Du et al., 2024), a binary mask matrix is generated by sampling a Bernoulli random variable with a predetermined mean for missing ratios. Additional missing mechanisms and their associated results are presented in Appendix D.

- **Baselines:** PSW-I is compared against representative TSI methods: (1) the predictive TSI methods (DLinear (Zeng et al., 2023), FreTS (Yi et al., 2023), TimesNet (Wu et al., 2023), iTransformer (Liu et al., 2024), PatchTST (Nie et al., 2023), Transformer(Vaswani et al., 2017), SAITS (Du et al., 2023) and SCINet (Liu et al., 2022a)), (2) the generative TSI methods (CSDI (Tashiro et al., 2021)). Additionally, the performance of distribution alignment methods tailored for non-temporal data (TDM (Zhao et al., 2023) and Sinkhorn (Muzellec et al., 2020)) is evaluated for comparison.

- **Implementation details:** To ensure consistency in experimental conditions, the batch size B is fixed at 256. The Adam optimizer, known for its adaptive update rate and effective convergence, is employed for training, with an update rate $\eta = 0.01$. We exclude 5% of the indices from the training data to form the validation set. The key hyperparameters involved in PSW-I are tuned to minimize the MSE on the validation set. The patch size is tuned within $\{24, 36, 48\}$; the matching strength is tuned within $\{1, 10, 100, 1000\}$. The experiments are conducted on a platform with two Intel(R) Xeon(R) Platinum 8383C CPUs @ 2.70GHz and a NVIDIA GeForce RTX 4090 GPU. Performance is evaluated using modified mean absolute error (MAE) and mean squared error (MSE) following (Zhao et al., 2023; Jarrett et al., 2022), with a focus on imputation errors over missing entries. We set $\text{T}_{\max} = 200$ and $\ell_{\max} = 1,000$ to ensure convergence, applying early stopping on the validation dataset with a patience of 10.

---

[1]The definition of the DFT matrix is presented in Definition C.4, and the gradient derivation is detailed in Theorem C.5. Notably, our gradient formulation omits the gradients from the optimal transport plan $\mathbf{T}$ with respect to $\alpha$ and $\beta$ to enhance the efficiency and stability of the calculation process.

Table 1: Imputation performance in terms of MSE and MAE on 10 datasets.

| Datasets | ETTh1 | | ETTh2 | | ETTm1 | | ETTm2 | | Electricity | | Traffic | | Weather | | Illness | | Exchange | | PEMS03 | |
|---|---|---|---|---|---|---|---|---|---|---|---|---|---|---|---|---|---|---|---|---|
| Metrics | MSE | MAE | MSE | MAE | MSE | MAE | MSE | MAE | MSE | MAE | MSE | MAE | MSE | MAE | MSE | MAE | MSE | MAE | MSE | MAE |
| Transformer | 0.222 | 0.322 | 0.221 | 0.312 | 0.06 | 0.16 | 0.041 | 0.134 | 0.12 | 0.228 | 0.216 | 0.214 | 0.195 | 0.132 | 0.24 | 0.3 | 0.224 | 0.186 | 0.081 | 0.184 |
| DLinear | 0.144 | 0.267 | 0.108 | 0.231 | 0.103 | 0.221 | 0.084 | 0.201 | 0.137 | 0.271 | 0.251 | 0.272 | 0.274 | 0.185 | 0.21 | 0.273 | 0.261 | 0.216 | 0.112 | 0.261 |
| TimesNet | 0.253 | 0.353 | 0.133 | 0.263 | 0.061 | 0.173 | 0.068 | 0.186 | 0.129 | 0.254 | 0.201 | 0.243 | 0.28 | 0.189 | 0.231 | 0.277 | 0.319 | 0.264 | 0.076 | 0.19 |
| FreTS | 0.184 | 0.312 | 0.147 | 0.259 | 0.055 | 0.159 | 0.039 | 0.135 | 0.155 | 0.285 | 0.234 | 0.27 | 0.178 | 0.12 | 0.278 | 0.325 | 0.228 | 0.189 | 0.109 | 0.252 |
| PatchTST | 0.171 | 0.297 | 0.126 | 0.258 | 0.05 | 0.149 | 0.03 | 0.118 | 0.138 | 0.262 | 0.235 | 0.238 | 0.247 | 0.167 | 0.605 | 0.505 | 0.237 | 0.197 | 0.065 | 0.179 |
| SCINet | 0.149 | 0.275 | 0.128 | 0.248 | 0.067 | 0.176 | 0.064 | 0.179 | 0.125 | 0.239 | 0.29 | 0.314 | 0.198 | 0.136 | 0.617 | 0.473 | 0.298 | 0.247 | 0.106 | 0.249 |
| iTransformer | 0.163 | 0.281 | 0.101 | 0.211 | 0.056 | 0.156 | 0.034 | 0.125 | 0.128 | 0.251 | 0.252 | 0.269 | 0.188 | 0.127 | 0.316 | 0.31 | 0.068 | 0.056 | 0.08 | 0.203 |
| SAITS | 0.216 | 0.305 | 0.183 | 0.256 | 0.056 | 0.154 | 0.042 | 0.129 | 0.114 | 0.216 | 0.224 | 0.207 | 0.132 | 0.089 | 0.167 | 0.216 | 1.005 | 0.833 | 0.083 | 0.189 |
| CSDI | 0.151 | 0.269 | 0.098 | 0.263 | 0.101 | 0.177 | 0.158 | 0.113 | 0.533 | 0.269 | 0.306 | 0.324 | 0.158 | 0.107 | 0.356 | 0.384 | 0.1 | 0.103 | 0.115 | 0.17 |
| Sinkhorn | 0.877 | 0.65 | 0.807 | 0.603 | 0.934 | 0.698 | 0.893 | 0.67 | 0.986 | 0.821 | 1.012 | 0.729 | 0.793 | 0.672 | 0.549 | 0.382 | 0.723 | 0.599 | 0.968 | 0.855 |
| TDM | 0.982 | 0.742 | 0.971 | 0.731 | 0.999 | 0.752 | 0.988 | 0.742 | 0.939 | 0.795 | 0.948 | 0.698 | 0.865 | 0.689 | 0.841 | 0.607 | 0.938 | 0.776 | 0.925 | 0.828 |
| **PSW-I (Ours)** | **0.126** | **0.231** | **0.046** | **0.142** | **0.047** | **0.131** | **0.021** | **0.094** | **0.106** | **0.208** | **0.197** | **0.199** | **0.107** | **0.072** | **0.067** | **0.122** | **0.031** | **0.026** | **0.049** | **0.149** |

*Kindly Note*: Each entry represents the average results at four missing ratios: 0.1, 0.3, 0.5, and 0.7. The best and second-best results are **bold** and underlined, respectively.

Table 2: Ablation study results.

| Model | SMR | PSD | Electricity | | ETTh1 | |
|---|---|---|---|---|---|---|
| | | | MSE | MAE | MSE | MAE |
| PSW-I[†] | ✗ | ✗ | $0.116_{\pm 0.004}$ | $0.227_{\pm 0.008}$ | $0.096_{\pm 0.002}$ | $0.204_{\pm 0.006}$ |
| PSW-I[‡] | ✗ | ✓ | $0.080^*_{\pm 0.003}$ | $0.196^*_{\pm 0.004}$ | $0.085^*_{\pm 0.001}$ | $0.208^*_{\pm 0.004}$ |
| PSW-I[††] | ✓ | ✗ | $0.115_{\pm 0.003}$ | $0.218_{\pm 0.005}$ | $0.091^*_{\pm 0.003}$ | $0.194^*_{\pm 0.006}$ |
| PSW-I | ✓ | ✓ | $\mathbf{0.075}^*_{\pm 0.003}$ | $\mathbf{0.180}^*_{\pm 0.004}$ | $\mathbf{0.077}^*_{\pm 0.002}$ | $\mathbf{0.188}^*_{\pm 0.004}$ |

*Kindly Note:* The best results are **bold**. "*" marks the results that significantly outperform PSW-I[†], with $p$-value $< 0.05$ over paired-sample t-test.

## 4.2 OVERALL PERFORMANCE

Table 1 compares the performance of PSW-I with baseline methods, averaged over four missing ratios: 0.1, 0.3, 0.5, and 0.7. Key observations from the results are as follows:

- **Effectiveness of Existing TSI Methods.** Existing TSI methods demonstrate promising performance. Notably, fundamental time-series models such as iTransformer and SCINet exhibit very competitive results, achieving the best performance among baselines in 4 out of 20 cases. These models effectively capture temporal patterns present in the data and leverage them for imputation tasks. Meanwhile, methods specifically crafted for TSI, such as SAITS and CSDI, also achieve comparable performance, each attaining the best performance among baselines in 4 out of 20 cases. Albeit with suboptimal structure for capturing temporal patterns, the specialized mechanisms for TSI, such as the conditional generation strategy in CSDI, effectively raise the performance comparable to advanced fundamental time-series models.

- **Limitations of Alignment-Based Methods in TSI.** Alignment-based methods, including Sinkhorn and TDM, demonstrate strong performance on non-temporal data (Zhao et al., 2023) but fall short in TSI tasks. Specifically, their performance in Table 1 is significantly inferior to that of prevailing TSI methods. This discrepancy can be attributed to their inability to capture temporal patterns inherent in time-series data and vulnerability to non-stationarity due to their inherent i.i.d. assumption (Muzellec et al., 2020), resulting in suboptimal imputation results.

- **Superiority of the Proposed PSW-I Method.** PSW-I retains the advantages of alignment-based methods: it does not require masking observed entries during training or training parametric models on incomplete data. This distinguishes it from prevailing TSI methods and positions it as a promising alternative. Moreover, PSW-I counteracts the limitation of alignment-based methods to capture temporal patterns and accommodate non-stationarity. Overall, PSW-I achieves the best performance across all 10 datasets, often surpassing the best baseline by significant margins (e.g., on PEMS03), showcasing its efficacy in real-world applications.

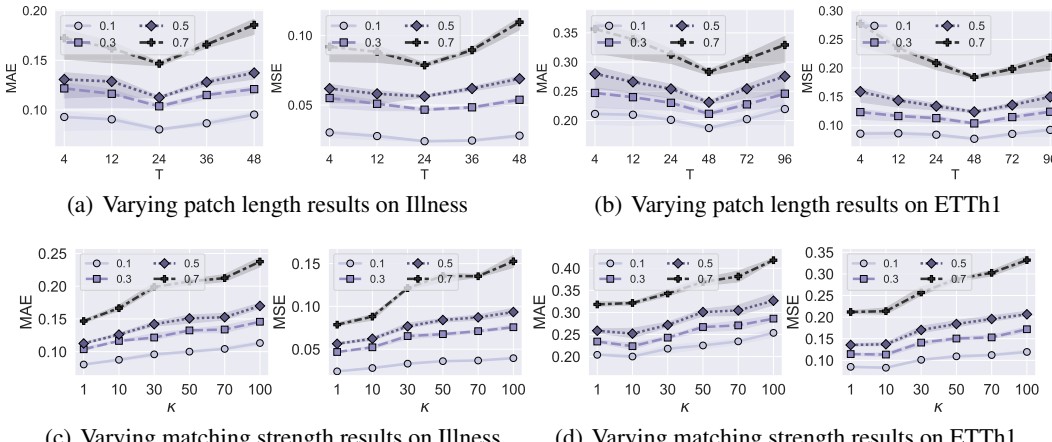

(a) Varying patch length results on Illness    (b) Varying patch length results on ETTh1

(c) Varying matching strength results on Illness    (d) Varying matching strength results on ETTh1

Figure 4: Varying patch length and matching strength results with missing ratios 0.1, 0.3, 0.5 and 0.7.

Table 3: Varying pairwise distance results.

| Distances | MSE | ΔMSE | MRE | ΔMRE | MAE | ΔMAE |
|---|---|---|---|---|---|---|
| | Electricity | | | | | |
| PSW-T | 0.080 | - | 0.233 | - | 0.193 | - |
| PSW-A | 0.079 | 1.3%↓ | 0.231 | 0.9%↓ | 0.191 | 1.0%↓ |
| PSW-P | 0.079 | 1.3%↓ | 0.229 | 1.7%↓ | 0.189 | 2.1%↓ |
| PSW | **0.075** | 6.7%↓ | **0.218** | 6.9%↓ | **0.180** | 7.2%↓ |
| | ETTh1 | | | | | |
| Distances | MAE | ΔMAE | MSE | ΔMSE | MRE | ΔMRE |
| PSW-T | 0.082 | - | 0.268 | - | 0.202 | - |
| PSW-A | 0.081 | 1.2%↓ | 0.265 | 1.1%↓ | 0.200 | 1.0%↓ |
| PSW-P | 0.081 | 1.2%↓ | 0.263 | 1.9%↓ | 0.198 | 2.0%↓ |
| PSW | **0.077** | 6.5%↓ | **0.250** | 7.2%↓ | **0.188** | 7.4%↓ |

Table 4: Varying discrepancy results.

| Discrepancies | MSE | ΔMSE | MRE | ΔMRE | MAE | ΔMAE |
|---|---|---|---|---|---|---|
| | Electricity | | | | | |
| OT | 0.082 | - | 0.239 | - | 0.196 | - |
| EMD | 0.080 | 2.5%↓ | 0.225 | 6.2%↓ | 0.184 | 6.5%↓ |
| UOT | 0.077 | 6.5%↓ | 0.226 | 5.8%↓ | 0.186 | 5.4%↓ |
| Ours | **0.075** | 9.3%↓ | **0.218** | 9.6%↓ | **0.180** | 8.9%↓ |
| | ETTh1 | | | | | |
| Discrepancies | MSE | ΔMSE | MRE | ΔMRE | MAE | ΔMAE |
| OT | 0.084 | - | 0.273 | - | 0.206 | - |
| EMD | 0.079 | 6.3%↓ | 0.264 | 3.4%↓ | 0.202 | 2.0%↓ |
| UOT | 0.081 | 3.7%↓ | 0.263 | 3.8%↓ | 0.198 | 4.0%↓ |
| Ours | **0.077** | 9.1%↓ | **0.250** | 9.2%↓ | **0.188** | 9.6%↓ |

## 4.3 ABLATIVE ANALYSIS

Table 2 presents an ablation study dissecting the contributions of the Pairwise Spectrum Distance (PSD) and Selective Matching Regularization (SMR), the two key components of the proposed PSW-I framework. The baseline model without PSD and SMR computes the Sinkhorn discrepancy using the Euclidean distance between patches. While this naive model performs suboptimally, it still outperforms the standard Sinkhorn imputation method (as shown in Table 1) because the patch-wise distance somewhat captures more temporal patterns than the step-wise distance.

Incorporating PSD and SMR significantly improves imputation performance by effectively encapsulating temporal patterns through spectral representation, and accommodating non-stationarity by matching a flexible set of samples. The PSW-I framework, which integrates both PSD and SMR, achieves the best performance, demonstrating the effectiveness of combining these two components.

## 4.4 GENERALITY ANALYSIS

In this section, we explore some alternative implementations to the key components of PSW-I to justify its rationale and advantages. Table 3 and 4 present the results, where Δ denotes the relative performance reduction. The primary observations are summarized as follows.

- **Effects of the Discrepancy Measure.** Our approach accommodates the non-stationarity by replacing the matching constraints in OT with a soft regularizer (SMR), leading to significant performance improvements over standard OT in Table 3. Notably, the UOT method in domain adaptation (Fatras et al., 2021; Séjourné et al., 2019) also relaxes matching constraints and can be adapted to TSI, similarly enhancing imputation performance compared to standard OT. However, our method differs from the standard UOT formulation by omitting entropic regularization, which proves to be beneficial to missing value imputation (see the discussion by Chen et al. (2024)). The results comparing OT (with entropic regularization) versus EMD (without entropic regularization)

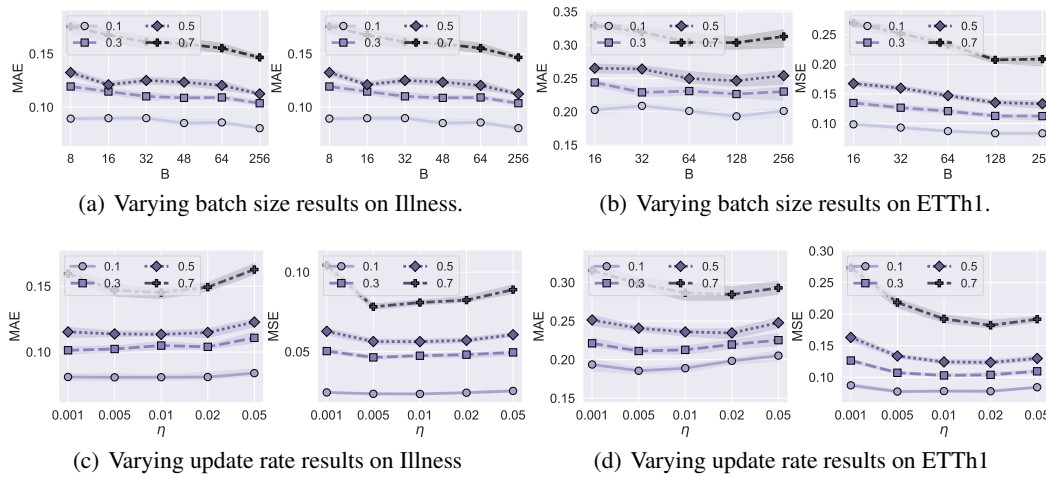

Figure 5: Varying batch size and update rate results with missing ratios 0.1, 0.3, 0.5 and 0.7.

are provided in Table 3. By avoiding entropic regularization, our method acquires rectified transport plans that are more effective for imputation tasks.

- **Effects of the Distance Metric.** The proposed Pairwise Spectrum Distance (PSD) computes the patch-wise distance by measuring the difference of their spectrum. Compared to calculating the patch-wise distance in the time domain (PSW-T in Table 3), using distance metrics that consider amplitude characteristics (PSW-A) or phase characteristics (PSW-P) individually leads to performance improvements, as they capture distinct aspects of temporal patterns. Specifically, PSW-A focuses on the strength of frequency components, while PSW-P captures the timing information inherent in the phase. Combining both amplitude and phase characteristics in PSW provides a comprehensive encapsulation of temporal patterns and achieves the best performance.

## 4.5 PARAMETER SENSITIVITY ANALYSIS

In this section, we examine the impact of critical hyperparameters on the performance of PSW-I. The results are presented in Fig. 4 and 5. The primary observations are summarized as follows:

- The patch length (T) determines the scale of temporal patterns that PSD captures. When T is reduced to 1, the model effectively degrades to naive alignment-based models (Muzellec et al., 2020) that overlook temporal patterns. Increasing T to 24 leads to substantial performance improvements, underscoring the importance of temporal patterns for TSI. However, further increasing T beyond 24 deteriorates performance, which could be attributed to the curse of dimensionality in the Wasserstein discrepancy, reducing its discriminability in high-dimensional spaces (Chizat et al., 2020).

- The matching strength ($\kappa$) controls the flexibility of matching mass in SMR. As $\kappa$ decreases, the robustness to non-stationarity property is enhanced. The consistent performance gains underscore the importance of accommodating non-stationarity for TSI and the efficacy of SMR.

- The batch size (B) determines the scale of the optimization problem when calculating PSW discrepancy. Overall, model performance is not highly sensitive to it in cases with low missing ratios. However, in cases with high missing ratios, a smaller batch size is beneficial, as it enables more fine-grained comparisons between distributions and can improve sample efficiency. Therefore, choosing a smaller batch size is advantageous for both accuracy and efficiency in such scenarios.

- The learning rate ($\eta$) controls the model convergence. As $\eta$ increases, the imputation error initially decreases and then increases, indicating the presence of an optimal value. A learning rate of 0.01 yields the best overall results, effectively balancing convergence stability and speed.

## 5 CONCLUSION

This study introduces the PSW-I approach, offering a fresh perspective on OT for time-series imputation. The core innovation is the PSW discrepancy: an instrumental and versatile discrepancy

measure for comparing distributions of time-series. Based on the proposed PSW discrepancy, we further derive a novel time-series imputation approach, termed PSW-I, which significantly enhances imputation performance on various real-world time-series datasets.

**Limitations and Future Work.** This work employs the DFT to encapsulate temporal patterns, which primarily captures global frequency components and may not effectively represent local or transient patterns. Future work could explore advanced spectral methods, such as wavelet transforms, to enhance temporal resolution. Additionally, due to the curse of dimensionality, there is a bottleneck when increasing the patch size to improve performance. Enhancing the scalability of PSW-I to exceedingly long-term patterns remains a challenging yet promising direction for future research.

## ACKNOWLEDGEMENT

This work was supported by National Natural Science Foundation of China (623B2002), National Science and Technology Major Project of China (2022ZD0120005), ARC DE210101624 and ARC DP240102088. The first author extends a special gratitude to Prof. Degui Yang of Central South University, for his exceptional signal processing lectures and generous research guidance during S.T.E.M. studies.

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

# A    RELATED WORKS

## A.1    MISSING VALUE IMPUTATION FOR TABULAR DATA

In this section, we review the predominant methodologies for missing data imputation in tabular datasets, which differ from time-series data due to the absence of temporal patterns and dependencies. Broadly, these methods can be categorized into four paradigms, each offering unique strengths and limitations (Jarrett et al., 2022).

The simple direct paradigm involves basic statistical methods such as mean, median, and mode imputation to replace missing values. These techniques are fast and easy to implement but often fall short in capturing the complex dependencies present in real-world datasets (Malarvizhi and Thanamani, 2012; Mazumder et al., 2010). As a result, this approach frequently leads to oversimplified and inadequate imputations that may not perform well in practice.

The round-robin paradigm (Stekhoven and Bühlmann, 2012; Royston and White, 2011), originally developed from the Iterative Conditional Expectation (ICE) algorithm (Royston and White, 2011), imputes missing values for each feature by utilizing other observable features as predictors. Building on ICE, modern approaches incorporate advanced parametric models, including neural networks (Mattei and Frellsen, 2019; Kyono et al., 2021), Bayesian models (Royston and White, 2011), and random forests (Stekhoven and Bühlmann, 2012), to better handle intricate missing patterns. Furthermore, diverse training strategies such as multiple imputation (Royston and White, 2011), ensemble learning (Stekhoven and Bühlmann, 2012), and multitask learning (Mattei and Frellsen, 2019) have been explored to improve performance across different contexts. Although this paradigm enhances flexibility and imputation accuracy, the necessity of careful model selection poses a challenge since the complete data is unavailable for validation (Jarrett et al., 2022).

The generative paradigm approaches imputation as a conditional generation problem, leveraging advanced neural architectures and generative models such as Generative Adversarial Networks (GANs) (Yoon et al., 2018; Sun et al., 2024; Li et al., 2024) and diffusion models (Tashiro et al., 2021; Xu et al., 2024; Ouyang et al., 2023; Chen et al., 2024) to estimate the underlying data distribution and generate plausible values for missing entries. This approach can capture complex relationships and dependencies, leading to high-quality imputations, especially when sufficient data is available. However, the generative paradigm inherits the challenges of generative models, including the instability of adversarial training and the significant data requirements of diffusion models. These issues are further exacerbated by the high missing ratios and complex missing patterns often encountered in industrial settings, limiting their practical utility.

A more recent innovation in this field is the distribution matching paradigm (Zhao et al., 2023; Muzellec et al., 2020; Wang et al., 2024b;a). This paradigm samples subsets of dataset and update the missing entries to minimize the discrepancy between the sampled subsets. The underlying assumption is that, under the i.i.d. principle, any two batches of data should follow the same distribution, allowing for effective imputation by reducing distributional divergence. Our research extends this methodology by addressing its vulnerability to outliers and enhancing its capability to account for temporal patterns, thus adapting it for time-series imputation tasks.

## A.2    MISSING VALUE IMPUTATION FOR TIME-SERIES DATA

Deep learning-based imputation methods have attracted substantial attention in TSI due to their ability to model complex nonlinearities and temporal dependencies in time-series. The literature can be broadly classified into two categories: predictive methods and generative methods (Du, 2023; Du et al., 2024; Wang et al., 2024c).

The predictive methods focus on estimating deterministic values for the missing entries within the time-series. These methods leverage various neural network architectures to capture nonlinearities and temporal dependencies, such as RNNs (e.g., DeepAR (Salinas et al., 2020), S4 (Gu et al., 2021), GRU-D (Che et al., 2018), BRITS (Cao et al., 2018)), CNNs (e.g., TimesNet (Wu et al., 2023)), and Transformers (e.g., iTransformer (Liu et al., 2024), SAITS (Du et al., 2023)). For example, GRU-D (Che et al., 2018) introduces a decay mechanism that effectively addresses irregularities arising from missing values, while TimesNet (Wu et al., 2023) transforms time-series data into an image-like representation using spectral analysis, allowing the application of vision models for imputation tasks.

Despite their ability to provide effective imputation, predictive methods are challenged by model selection amidst incomplete data. Additionally, these models typically require masking of some observed entries during training to generate labels, which can reduce sample efficiency, reducing performance especially when the proportion of missing values is high.

The generative methods aim to generate missing values conditionally given observed ones (Chen et al., 2023; Luo et al., 2018; Gao et al., 2024). Notable generative approaches include variational autoencoders (VAEs), generative adversarial networks (GANs), and diffusion models. For instance, Fortuin et al. (2020) proposed GP-VAE, a model that integrates a Gaussian process prior with a variational autoencoder to model incomplete time-series data. Similarly, Miao et al. (2021) introduced US-GAN, which improves the masking matrix to facilitate the generator during imputation. Tashiro et al. (2021) proposed CSDI, a diffusion model that employs a conditioned training strategy to guide the imputation of missing entries. While generative methods offer a more probabilistic framework and can capture the inherent uncertainty in the imputation process, they are often more computationally intensive and complex to train, which can present scalability issues. Similar to predictive methods, generative models also require masking observed entries to construct training labels, leading to reduced sample efficiency, particularly in cases with a high proportion of missing data.

## B    BACKGROUND ON DISCRETE OPTIMAL TRANSPORT

This section introduces the foundational concepts and algorithms necessary for computing OT between discrete measures. Consider a scenario involving $n$ warehouses and $m$ factories, where the $i$-th warehouse holds $\mathbf{a}_i$ units of material and the $j$-th factory requires $\mathbf{b}_j$ units of material (Peyré and Cuturi, 2019). The objective is to establish a mapping from warehouses to factories that: (1) completely allocates all warehouse materials, (2) fulfills all factory demands, and (3) restricts the transportation from each warehouse to at most one factory. Each potential mapping is evaluated based on a global cost, which aggregates the local costs incurred from transporting a unit of material from warehouse $i$ to factory $j$.

**Definition B.1.** *The Monge problem for discrete measures, where $\alpha = \sum_{i=1}^{n} \mathbf{a}_i \delta_{\mathbf{x}_i}$ and $\beta = \sum_{j=1}^{m} \mathbf{b}_j \delta_{\mathbf{x}_j}$, seeks a mapping $\mathbb{T} : \{\mathbf{x}_i\}_{i=1}^{n} \to \{\mathbf{x}_j\}_{j=1}^{m}$ that optimally redistributes the mass from $\alpha$ to $\beta$. Specifically, for each $j$, it must hold that $\mathbf{b}_j = \sum_{i:\mathbb{T}(\mathbf{x}_i)=\mathbf{x}_j} \mathbf{a}_i$ and $\mathbb{T}_\sharp \alpha = \beta$. The goal is to minimize the transportation cost, represented by $c(x, y)$, leading to the following formulation:*

$$\min_{\mathbb{T}:\mathbb{T}_\sharp\alpha=\beta} \left\{ \sum_i c(\mathbf{x}_i, \mathbb{T}(\mathbf{x}_i)) \right\}.$$

The original Monge formulation does not guarantee the existence or uniqueness of solutions (Peyré and Cuturi, 2019). Therefore, Kantorovich (Kantorovich, 2006) extended this framework by relaxing the one-to-one mapping constraint, allowing transportation from a single warehouse to multiple factories, and reformulated the problem as a linear programming problem.

**Definition B.2.** *The Kantorovich problem for discrete measures $\alpha$ and $\beta$ defines a cost-minimization task over feasible transport plans $\pi \in \mathbb{R}_+^{n \times m}$. The objective is to find a plan that minimizes the overall transport cost:*

$$\mathcal{W}(\alpha, \beta) := \min_{\pi \in \Pi(\alpha,\beta)} \langle \mathbf{D}, \pi \rangle,$$

$$\Omega(\alpha, \beta) := \left\{ \pi \in \mathbb{R}_+^{n \times m} : \pi \mathbf{1}_\mathrm{m} = \mathbf{a}, \pi^\mathrm{T} \mathbf{1}_\mathrm{n} = \mathbf{b} \right\},$$

*where $\mathcal{W}(\alpha, \beta)$ represents the Wasserstein discrepancy between $\alpha$ and $\beta$; $\mathbf{D}$ denotes the distance matrix computed using the squared Euclidean metric (Courty et al., 2017), and $\mathbf{a}$, $\mathbf{b}$ are vectors describing the mass distribution in $\alpha$ and $\beta$, respectively.*

The subsequent research in discrete OT primarily follows two trajectories. The first aims to reduce the computational complexity of solving OT problems. While exact solutions can be obtained through linear programming algorithms, these come with cubic complexity in relation to the number of samples (Bonneel et al., 2011). To address this, various approximate algorithms for acceleration have been developed, such as the Sinkhorn and sliced OT algorithm (Altschuler et al., 2017) with quadratic and linear complexity, respectively. The second line of research focuses on modifying the transport

---

**Algorithm 1** The imputation workflow of PSW-I.

---

**Input**: $\mathbf{X}^{(\mathrm{obs})}$: the incomplete data; $\mathbf{M}$: the mask matrix.
**Parameter**: $\kappa$: the matching strength; $\varepsilon$ the entropic regularization strength; $\ell_{\max}$: the max number of iterations to solve PSW; $\mathrm{T}_{\max}$: the max number of epochs; B: the batch size; $\eta$: the update rate.
**Output**: $\mathbf{X}^{(\mathrm{imp})}$: the imputed dataset.

1: $t \leftarrow 0$.
2: $\mathbf{X}^t \leftarrow \mathrm{Pre\text{-}Impute}(\mathbf{X}^{(\mathrm{obs})})$
3: **while** $t < \mathrm{T}_{\max}$ **do**
4:     $\alpha, \beta, \mathbf{M}^\alpha, \mathbf{M}^\beta \leftarrow \mathrm{Sample}(\mathbf{X}^t, \mathbf{M}; \mathrm{B})$.
5:     $\alpha^{(\mathrm{F})} \leftarrow \mathcal{F}(\alpha), \quad \beta^{(\mathrm{F})} \leftarrow \mathcal{F}(\beta)$
6:     $\mathbf{D}_{ij}^{(\mathrm{F})} = \|\alpha_i^{(\mathrm{F})} - \beta_j^{(\mathrm{F})}\|_1, \quad 1 \leq i, j \leq \mathrm{B}$
7:     $\mathcal{P} \leftarrow \mathrm{Algorithm1}(\mathbf{D}^{(\mathrm{F})}; \varepsilon, \kappa, \ell_{\max})$.
8:     $\alpha' \leftarrow \alpha - \eta \nabla_\alpha \mathcal{P} \odot \mathbf{M}$.
9:     $\beta' \leftarrow \beta - \eta \nabla_\beta \mathcal{P} \odot \mathbf{M}$.
10:     $\mathbf{X}^{t+1} \leftarrow \mathrm{Update}(\alpha', \beta', \mathbf{X}^t)$.
11:     $t \leftarrow t + 1$.
12:     **if** $\|\mathbf{X}^t - \mathbf{X}^{t-1}\|_{\mathrm{F}} < 1e^{-4}$ **then**
13:         **break**.
14: $\mathbf{X}^{(\mathrm{imp})} \leftarrow \mathbf{X}^t$.

---

problem to suit specific applications. Examples include the weak transport problem in domain adaptation (Chizat et al., 2018), the Schrödinger bridge problem in generative modeling (Marino and Gerolin, 2020), the Gromov problem in graph matching (Xu et al., 2019), the unbalanced transport problem in causal inference (Wang et al., 2023), and the adversarial formulation in GANs (Yoon et al., 2018; Spinelli et al., 2020; Li et al., 2024).

## C   THEORETICAL ANALYSIS

### C.1   THEORETICAL JUSTIFICATIONS

**Theorem C.1** (Metric Properties). *PSW defines a valid divergence on the space of probability measures over time series. For any probability measures $\alpha, \beta \in \mathbb{R}^T$, PSW satisfies:*

*(i) Non-negativity: $\mathcal{P}(\alpha, \beta) \geq 0$,*

*(ii) Identity of Indiscernibles: $\mathcal{P}(\alpha, \beta) = 0$ if and only if $\alpha = \beta$,*

*(iii) Symmetry: $\mathcal{P}(\alpha, \beta) = \mathcal{P}(\beta, \alpha)$.*

*Moreover, when $\kappa = 0$, PSW satisfies the triangle inequality and thus constitutes a metric.*

*Proof.* The proof proceeds by verifying each of the four metric properties for the PSW discrepancy.

**Non-Negativity:** The PSW discrepancy is defined via an optimal transport formulation incorporating the spectral distance metric $d_{\mathrm{spectral}}$, the transport strategy, and a selective matching regularizer. All these components are non-negative. Consequently, $\mathcal{P}(\alpha, \beta)$—the infimum of a sum of non-negative terms—satisfies $\mathcal{P}(\alpha, \beta) \geq 0$.

**Identity of Indiscernibles:** If $\alpha = \beta$, the optimal coupling $\gamma$ is the identity coupling, and $d_{\mathrm{spectral}}(\alpha_i, \beta_j) = 0$ for $i, j$ such that $\gamma_{i,j} > 0$. The selective mass regularizer contributes zero or a minimal constant that can be canceled based on its formulation. Thus, $\mathcal{P}(\alpha, \beta) = 0$. Conversely, if $\mathcal{P}(\alpha, \beta) = 0$, the spectral distance metric $d_{\mathrm{spectral}}(x, y)$ must vanish almost surely under the optimal coupling $\gamma$. Assuming the DFT is injective for sufficiently rich and non-degenerate time series, we have $x = y$ almost everywhere, leading to $\alpha = \beta$.

**Symmetry:** The spectral distance metric satisfies $d_{\mathrm{spectral}}(x, y) = d_{\mathrm{spectral}}(y, x)$. Therefore, swapping $\alpha$ and $\beta$ does not change $\mathcal{P}(\alpha, \beta)$. Moreover, transposing $\gamma$ does not change the value of

selective mass regularizer, which immediately follows from the common assumption that the sample masses are uniformly distributed in each distribution, ensuring $\mathcal{P}(\alpha, \beta) = \mathcal{P}(\beta, \alpha)$.

**Triangle Inequality (when $\kappa = 0$):** Consider three probability distributions $\alpha$, $\beta$, and $\xi$. Let $\gamma_{\alpha,\beta} \in \gamma(\alpha, \beta)$ and $\gamma_{\beta,\xi} \in \gamma(\beta, \xi)$ be the optimal couplings that minimize $\mathcal{P}(\alpha, \beta)$ and $\mathcal{P}(\beta, \xi)$, respectively. Utilizing the concept of *transitivity* in optimal transport, define the coupling $\gamma_{\alpha,\xi}$ by

$$\gamma_{\alpha,\xi}(x, z) = \int \gamma_{\alpha,\beta}(x, y)\gamma_{\beta,\xi}(y, z)\, dy.$$

Next, we compare the PSW discrepancy between $\alpha$ and $\xi$ with the discrepancy based on $\gamma_{\alpha,\xi}$:

$$\mathcal{P}(\alpha, \xi) \leq \int d_{\text{spectral}}(x, z)\, d\gamma_{\alpha,\xi}(x, z).$$

Substituting the expression for $\gamma_{\alpha,\xi}$, we have:

$$\mathcal{P}(\alpha, \xi) \leq \int d_{\text{spectral}}(x, z) \left( \int \gamma_{\alpha,\beta}(x, y)\gamma_{\beta,\xi}(y, z)\, dy \right) dx\, dz.$$

Applying Fubini's theorem to interchange the order of integration, the above expression becomes:

$$\mathcal{P}(\alpha, \xi) \leq \int \left( \int d_{\text{spectral}}(x, z)\gamma_{\beta,\xi}(y, z)\, dz \right) \gamma_{\alpha,\beta}(x, y)\, dy\, dx.$$

By the triangle inequality property of the spectral distance metric $d_{\text{spectral}}$, we have:

$$d_{\text{spectral}}(x, z) \leq d_{\text{spectral}}(x, y) + d_{\text{spectral}}(y, z).$$

Substituting this into the integral, we obtain:

$$\mathcal{P}(\alpha, \xi) \leq \int \left( d_{\text{spectral}}(x, y) + d_{\text{spectral}}(y, z) \right) \gamma_{\alpha,\beta}(x, y)\gamma_{\beta,\xi}(y, z)\, dy\, dx\, dz.$$

This expression can be separated into two distinct integrals:

$$\mathcal{P}(\alpha, \xi) \leq \int d_{\text{spectral}}(x, y)\gamma_{\alpha,\beta}(x, y)\, dy\, dx + \int d_{\text{spectral}}(y, z)\gamma_{\beta,\xi}(y, z)\, dy\, dz.$$

Recognizing that the first integral corresponds to $\mathcal{P}(\alpha, \beta)$ and the second to $\mathcal{P}(\beta, \xi)$, we have:

$$\mathcal{P}(\alpha, \xi) \leq \mathcal{P}(\alpha, \beta) + \mathcal{P}(\beta, \xi).$$

Thus, $\mathcal{P}$ satisfies the triangle inequality when $\kappa = 0$.

$\square$

**Theorem C.2** (Robustness of PSW to Outlier modes). *Suppose that $\tilde{\alpha} = \zeta\delta_z + (1 - \zeta)\alpha$ is a distribution disturbed by a Dirac mode at $z$ with relative mass $\zeta \in (0, 1)$. We have:*

$$\mathcal{P}^\kappa(\tilde{\alpha}, \beta) \leq (1 - \zeta)\mathcal{P}^\kappa(\alpha, \beta) + 2\kappa\zeta(1 - e^{-d(z)/2\kappa}).$$

*where $d(z) = \left\langle \mathbf{D}^{(\mathrm{F})}(z, \beta), \Delta_{\mathrm{m}} \right\rangle$ is the average distance of $z$ and samples in $\beta$.*

*Proof.* This theorem builds upon the foundation of Lemma 1.1 by Fatras et al. (2021) and extends its applicability to our proximal spectrum wasserstein discrepancy. Let $\mathbf{T}^*$ be the OT plan associated with $\mathcal{P}^\kappa(\alpha, \beta)$ in (2). For the disturbed distribution $\tilde{\alpha}$, suppose the transport plan is $\tilde{\mathbf{T}}^* = (1 - \zeta)\mathbf{T}^* + \zeta\tilde{\delta}_z \times \beta$, where $\phi$ is a parameter to be optimized. The marginals of $\tilde{\mathbf{T}}^*$ are $\tilde{\mathbf{T}}_1^* = (1 - \zeta)\mathbf{T}_1^* + \zeta\phi\delta_z$ and $\tilde{\mathbf{T}}_2^* = (1 - \zeta)\mathbf{T}_2^* + \zeta\phi\beta$. The KL divergence of them to $\alpha$ and $\beta$ satisfies

$$\mathrm{D_{KL}}(\tilde{\mathbf{T}}_1^*\|\tilde{\alpha}) \leq (1 - \zeta)\mathrm{D_{KL}}(\mathbf{T}_1^*\|\alpha) + \zeta\mathrm{D_{KL}}(\phi\delta_z\|\delta_z)$$

$$\mathrm{D_{KL}}(\tilde{\mathbf{T}}_2^*\|\beta) \leq (1 - \zeta)\mathrm{D_{KL}}(\mathbf{T}_2^*\|\beta) + \zeta\mathrm{D_{KL}}(\phi\beta\|\beta)$$

which immediately follows from the joint convexity of KL divergence. Therefore, we have:

$$\mathcal{P}^\kappa(\tilde{\alpha}, \beta) \leq (1 - \zeta)\overbrace{\left\langle \mathbf{D}^{(\mathrm{F})}, \mathbf{T}^* \right\rangle + \kappa\mathrm{D_{KL}}(\mathbf{T}_1^*\|\alpha) + \kappa\mathrm{D_{KL}}(\mathbf{T}_2^*\|\beta)}^{\mathcal{P}^\kappa(\alpha,\beta)}$$

$$+ \zeta\overbrace{\left[ \phi \left\langle \mathbf{D}^{(\mathrm{F})}(z, \beta), \Delta_{\mathrm{m}} \right\rangle + \mathrm{D_{KL}}(\phi\delta_z\|\delta_z) + \mathrm{D_{KL}}(\phi\beta\|\beta) \right]}^{e(\phi)},$$

where $d(z) = \langle \mathbf{D}^{(\mathrm{F})}(z, \beta), \Delta_{\mathrm{m}} \rangle$ can be viewed as the average from the distance of $z$ and samples in $\beta$. According to the first order condition, the term $e(\phi)$ achieves the minimum when $\phi = e^{-d(z)/2\kappa}$. Substituting it into the above inequality, we have:

$$\mathcal{P}^\kappa(\tilde{\alpha}, \beta) \le (1 - \zeta)\mathcal{P}^\kappa(\alpha, \beta) + 2\kappa\zeta(1 - e^{-d(z)/2\kappa}).$$

$\square$

**Theorem C.3** (Error Bounds of PSW-I). *Suppose $\theta_k = (\alpha_k, \beta_k)$ is the imputation values at the $k$-th iteration, and $\theta^* = (\alpha^*, \beta^*)$ be the optimum imputation values that minimizes $\mathcal{P}(\alpha, \beta)$. Assume that $\mathcal{P}$ satisfies the following regularity conditions:*

*1. Convexity: $\mathcal{P}(\theta)$ is convex with respect to $(\theta)$, which naturally holds since it is an composition of affine functions and 1-norms.*

*2. Smoothness: $\mathcal{P}(\theta)$ has Lipschitz continuous gradients with Lipschitz constant $L > 0$, i.e.,*

$$\|\nabla_\theta \mathcal{P}(\theta_k) - \nabla_\theta \mathcal{P}(\theta_{k+1})\| \le L\|\theta_{k+1} - \theta_k\|,$$

*3. Initialization Proximity: The initial distributions satisfy $\|\theta_0 - \theta^*\| \le \epsilon$ for some constant $\epsilon > 0$.*

*Under these conditions, for all iterations $\mathrm{K} \ge 0$, the PSW-I framework ensures:*

$$\mathcal{P}(\alpha_k, \beta_k) - \mathcal{P}(\alpha^*, \beta^*) \le \frac{1}{2\eta\mathrm{K}}\epsilon^2.$$

*Proof.* Given that $\mathcal{P}(\alpha, \beta)$ is convex with respect to $(\alpha, \beta)$ and has $L$-Lipschitz continuous gradients, we can follow the standard steps for deriving the error bound for optimizations.

Suppose $\theta_k = (\alpha_k, \beta_k)$ is the imputation values at the $k$-th iteration, and $\theta^* = (\alpha^*, \beta^*)$ be the optimum imputation values that minimizes $\mathcal{P}(\alpha, \beta)$. Consider the iterative update rules of the PSW-I framework using gradient descent:

$$\theta_{k+1} = \theta_k - \eta\nabla_\theta \mathcal{P}(\theta_k),$$

For a L-Lipschitz convex function $\mathcal{P}$, we have

$$
\begin{aligned}
\mathcal{P}(\theta_{k+1}) &\le \mathcal{P}(\theta_k) + \nabla_\theta \mathcal{P}(\theta_k)(\theta_{k+1} - \theta_k) + \frac{L}{2}\|\theta_{k+1} - \theta_k\|^2 \\
&= \mathcal{P}(\theta_k) - \eta\|\nabla_\theta \mathcal{P}(\theta_k)\|^2 + \frac{L\eta^2}{2}\|\nabla_\theta \mathcal{P}(\theta_k)\|^2 \\
&= \mathcal{P}(\theta_k) - \eta(1 - \frac{L\eta}{2})\|\nabla_\theta \mathcal{P}(\theta_k)\|^2 \\
&\le \mathcal{P}(\theta_k) - \frac{\eta}{2}\|\nabla_\theta \mathcal{P}(\theta_k)\|^2 \\
&\le \mathcal{P}(\theta^*) + \nabla_\theta \mathcal{P}(\theta_k)(\theta_k - \theta^*) - \frac{\eta}{2}\|\nabla_\theta \mathcal{P}(\theta_k)\|^2 \\
&= \mathcal{P}(\theta^*) + \frac{1}{2\eta}\|\theta_k - \theta^*\|^2 - \frac{1}{2\eta}\|\theta_k - \theta^*\|^2 + \nabla_\theta \mathcal{P}(\theta_k)(\theta_k - \theta^*) - \frac{\eta}{2}\|\nabla_\theta \mathcal{P}(\theta_k)\|^2 \\
&= \mathcal{P}(\theta^*) + \frac{1}{2\eta}\|\theta_k - \theta^*\|^2 - \frac{1}{2\eta}\|\theta_k - \theta^* - \eta\nabla_\theta \mathcal{P}(\theta_k)\|^2 \\
&= \mathcal{P}(\theta^*) + \frac{1}{2\eta}(\|\theta_k - \theta^*\|^2 - \|\theta_{k+1} - \theta^*\|^2),
\end{aligned}
$$

(3)

where the first line is a well-known property of Lipschitz convex functions; the second line incorporates the update rules of the gradient descent, the fourth line holds as we set $\eta \le \frac{1}{L}$ which makes $-\eta + \frac{L\eta^2}{2} \le -\frac{\eta}{2}$.

Summing up the inequality above with $k = 0, 1, ..., \mathrm{K}$:

$$\sum_{k=1}^{\mathrm{K}} \mathcal{P}(\theta_k) - \mathrm{K}\mathcal{P}(\theta^*) \le \frac{1}{2\eta}(\|\theta_0 - \theta^*)\|^2 - \|\theta_{\mathrm{K}} - \theta^*\|^2) \le \frac{1}{2\eta}\|\theta_0 - \theta^*\|^2.$$

According to the 4-th line of (3), we have $\mathcal{P}(\theta_{k+1}) \le \mathcal{P}(\theta_k) \le \dots \le \mathcal{P}(\theta_0)$. Therefore, $\mathrm{K}\mathcal{P}(\theta_\mathrm{K}) \le \sum_{k=1}^\mathrm{K} \mathcal{P}(\theta_k)$ holds, which immediately follows by

$$\mathrm{K}\mathcal{P}(\theta_\mathrm{K}) - \mathrm{K}\mathcal{P}(\theta^*) \le \sum_{k=1}^\mathrm{K} \mathcal{P}(\theta_k) - \mathrm{K}\mathcal{P}(\theta^*) \le \frac{1}{2\eta}\|\theta_0 - \theta^*\|^2 \le \frac{1}{2\eta}\epsilon^2.$$

Therefore, we have:

$$\mathcal{P}(\theta_\mathrm{K}) - \mathcal{P}(\theta^*) \le \frac{1}{2\eta\mathrm{K}}\epsilon^2.$$

$\square$

**Definition C.4** (DFT). Let $\mathbf{x} = [x_0, x_1, \dots, x_{\mathrm{T}-1}]$ denote a T-length sequence, the DFT of $\mathbf{x}$ is a T-length sequence with the $k$-th component defined as:

$$\mathbf{x}_k^{(\mathrm{F})} = \sum_{t=0}^{\mathrm{T}-1} \mathbf{x}_t e^{-j2\pi kt/\mathrm{T}} = [\mathbf{x}_0, .., \mathbf{x}_{\mathrm{T}-1}] \cdot \begin{bmatrix} \mathbf{W}_{0,k}^{(\mathrm{F})} \\ \mathbf{W}_{1,k}^{(\mathrm{F})} \\ \\ \mathbf{W}_{n-1,k}^{(\mathrm{F})} \end{bmatrix}$$

where $j$ is the imaginary unit, $\mathbf{W}_{t,k}^{(\mathrm{F})} := e^{-j2\pi kt/\mathrm{T}}$. On the basis, the DFT sequence $\mathbf{x}^{(\mathrm{F})} = [\mathbf{x}_0^{(\mathrm{F})}, .., \mathbf{x}_{\mathrm{T}-1}^{(\mathrm{F})}]$ can be calculated as:

$$\mathbf{x}^{(\mathrm{F})} = [\mathbf{x}_0, .., \mathbf{x}_{n-1}] \cdot \begin{bmatrix} \mathbf{W}_{0,0}^{(\mathrm{F})} & \mathbf{W}_{0,1}^{(\mathrm{F})} & \cdots & \mathbf{W}_{0,\mathrm{T}-1}^{(\mathrm{F})} \\ \mathbf{W}_{1,0}^{(\mathrm{F})} & \mathbf{W}_{1,1}^{(\mathrm{F})} & \cdots & \mathbf{W}_{1,\mathrm{T}-1}^{(\mathrm{F})} \\ \vdots & \vdots & \ddots & \vdots \\ \mathbf{W}_{\mathrm{T}-1,0}^{(\mathrm{F})} & \mathbf{W}_{\mathrm{T}-1,1}^{(\mathrm{F})} & \cdots & \mathbf{W}_{\mathrm{T}-1,\mathrm{T}-1}^{(\mathrm{F})} \end{bmatrix} = \mathbf{x} \cdot \mathbf{W}^{(\mathrm{F})}$$

where $\mathbf{W}^{(\mathrm{F})}$ is defined as the DFT matrix.

**Theorem C.5** (Gradient calculation). *The gradients of the PSW discrepancy with respect to $\alpha$ and $\beta$ can be expressed as*

$$\frac{\partial\mathcal{P}}{\partial\alpha_i} = \sum_{j=1}^\mathrm{m} \mathbf{T}_{i,j}\mathbf{W}^{(\mathrm{F})} \cdot \mathrm{sign}(\mathbf{e}_{i,j})^\top,$$

$$\frac{\partial\mathcal{P}}{\partial\beta_j} = -\sum_{i=1}^\mathrm{n} \mathbf{T}_{i,j}\mathbf{W}^{(\mathrm{F})} \cdot \mathrm{sign}(\mathbf{e}_{i,j})^\top.$$

*where the gradient of $\mathbf{T}$ to $\alpha$ and $\beta$ is truncated for accelerating and stabilizing the update process.*

*Proof.* After acquiring the optimal transport matrix–$\mathbf{T}$–by solving the optimization problem in (2), the PSW is calculated as

$$\mathcal{P}(\alpha, \beta) := \left\langle \mathbf{D}^{(\mathrm{F})}, \mathbf{T} \right\rangle + \kappa \left( \mathrm{D}_{\mathrm{KL}}(\mathbf{T}\mathbf{1}_m\|\Delta_n) + \mathrm{D}_{\mathrm{KL}}(\mathbf{T}^\mathrm{T}\mathbf{1}_n\|\Delta_m) \right)$$

which depends on $\alpha$ and $\beta$ through the distance matrix $\mathbf{D}_{i,j}^{(\mathrm{F})} = \|\alpha_i^{(\mathrm{F})} - \beta_j^{(\mathrm{F})}\|_1 = \|(\alpha_i - \beta_j)\mathbf{W}^{(\mathrm{F})}\|_1$, where $\mathbf{W}^{(\mathrm{F})}$ is the DFT matrix in Definition C.4. Using the chain rule, we have:

$$\frac{\partial\mathcal{P}}{\partial\alpha_i} = \sum_{j=1}^\mathrm{m} \frac{\partial\mathcal{P}}{\partial\mathbf{D}_{i,j}^{(\mathrm{F})}} \frac{\partial\mathbf{D}_{i,j}^{(\mathrm{F})}}{\partial\alpha_i} = \sum_{j=1}^\mathrm{m} \mathbf{T}_{i,j} \frac{\partial\mathbf{D}_{i,j}^{(\mathrm{F})}}{\partial\alpha_i} = \sum_{j=1}^\mathrm{m} \mathbf{T}_{i,j}\mathbf{W}^{(\mathrm{F})} \cdot \mathrm{sign}(\mathbf{e}_{i,j})^\top,$$

$$\frac{\partial\mathcal{P}}{\partial\beta_j} = \sum_{i=1}^\mathrm{n} \frac{\partial\mathcal{P}}{\partial\mathbf{D}_{i,j}^{(\mathrm{F})}} \frac{\partial\mathbf{D}_{i,j}^{(F)}}{\partial\beta_j} = \sum_{i=1}^\mathrm{n} \mathbf{T}_{i,j} \frac{\partial\mathbf{D}_{i,j}^{(\mathrm{F})}}{\partial\beta_j} = -\sum_{i=1}^\mathrm{n} \mathbf{T}_{i,j}\mathbf{W}^{(\mathrm{F})} \cdot \mathrm{sign}(\mathbf{e}_{i,j})^\top.$$

where $\mathbf{e}_{i,j} = (\alpha_i - \beta_j)\mathbf{W}^{(\mathrm{F})} \in \mathbb{R}^\mathrm{T}$, T is the sequence length. Notably, the derivatives $d\mathbf{T}/d\alpha_i$ and $d\mathbf{T}/d\beta_j$ should be considered ideally. However, $\mathbf{T}$ is obtained through an iterative numerical optimization procedure that is not always differentiable. Even if the optimization procedure was

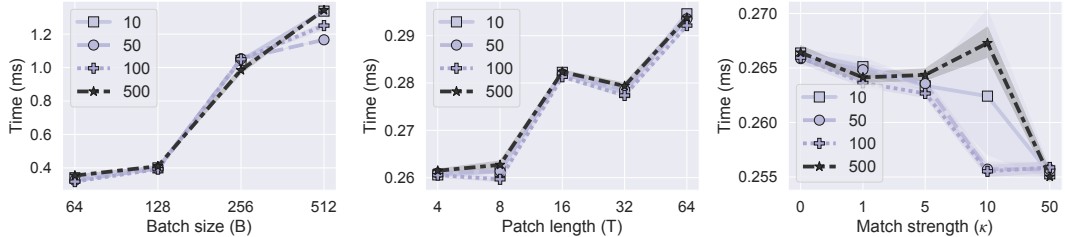

Figure 6: Running time of calculating PSW under diverse settings. Each color indicates a distinct maximum of iterations. The colored lines indicate the mean values of 100 trials, with the shadowed areas representing 99.9% confidence intervals.

differentiable, calculating $d\mathbf{T}/d\alpha_i$ and $d\mathbf{T}/d\beta_j$ would be computationally intensive due to the need to compute gradients across up to 1,000 iteration steps. Even worse, large values in iteration could cause excessively large gradients and lead to gradient explosion. Therefore, we choose to update $\alpha$ and $\beta$ through the pathway $\mathcal{P} \to \mathbf{D}^{(\mathrm{F})} \to \alpha, \beta$. This method stabilizes the optimization procedure and effectively guides the imputation updates.

Finally, let $\mathbf{e}_{i,j,k}$ be the $k$-th element of $\mathbf{e}_{i,j}$, we detail the derivation of $\frac{\partial \mathbf{D}_{i,j}^{(\mathrm{F})}}{\partial \alpha_i}$ and $\frac{\partial \mathbf{D}_{i,j}^{(\mathrm{F})}}{\partial \beta_j}$ as follows:

$$
\begin{aligned}
\frac{\partial \mathbf{D}_{i,j}^{(\mathrm{F})}}{\partial \alpha_i} &= \sum_k \frac{\partial \mathbf{D}_{i,j}^{(\mathrm{F})}}{\mathbf{e}_{i,j,k}} \frac{\partial \mathbf{e}_{i,j,k}}{\partial \alpha_i} \\
&= \sum_k \mathrm{sign}(\mathbf{e}_{i,j,k}) \frac{\partial \mathbf{e}_{i,j,k}}{\partial \alpha_i} \\
&= \sum_k \mathrm{sign}(\mathbf{e}_{i,j,k}) \mathbf{W}_{:,k}^{(\mathrm{F})} = [\mathbf{W}_{:,0}^{(\mathrm{F})}, ..., \mathbf{W}_{:,\mathrm{n}-1}^{(\mathrm{F})}][\mathrm{sign}(\mathbf{e}_{i,j,0}), ..., \mathrm{sign}(\mathbf{e}_{i,j,\mathrm{n}-1})]^\top \\
&= \mathbf{W}^{(\mathrm{F})} \mathrm{sign}(\mathbf{e}_{i,j})^\top, \\
\frac{\partial \mathbf{D}_{i,j}^{(\mathrm{F})}}{\partial \beta_j} &= \sum_k \frac{\partial \mathbf{D}_{i,j}^{(\mathrm{F})}}{\mathbf{e}_{i,j,k}} \frac{\partial \mathbf{e}_{i,j,k}}{\partial \alpha_i} \\
&= \sum_k \mathrm{sign}(\mathbf{e}_{i,j,k}) \frac{\partial \mathbf{e}_{i,j,k}}{\partial \beta_j} \\
&= -\sum_k \mathrm{sign}(\mathbf{e}_{i,j,k}) \mathbf{W}_{:,k}^{(\mathrm{F})} = -[\mathbf{W}_{:,0}^{(\mathrm{F})}, ..., \mathbf{W}_{:,\mathrm{n}-1}^{(\mathrm{F})}][\mathrm{sign}(\mathbf{e}_{i,j,0}), ..., \mathrm{sign}(\mathbf{e}_{i,j,\mathrm{n}-1})]^\top \\
&= -\mathbf{W}^{(\mathrm{F})} \mathrm{sign}(\mathbf{e}_{i,j})^\top.
\end{aligned}
$$

$\square$

# D  ADDITIONAL EXPERIMENTS

## D.1  RUNNING TIME ANALYSIS

We examine the computational demands of the proposed PSW-I method, focusing on the time required for each update, specifically the execution time of Algorithm 1 to calculate the PSW distance. Results are detailed in Fig. 6 and delineated below.

- Increasing B from 64 to 512, while keeping T = 8, leads to a predictable increase in running time. This is due to the enlargement of the transport matrix ($\mathbf{T}$), which escalates the complexity of the optimization problem. Nevertheless, the running time remains manageable, with durations near 1.2 seconds even at the largest batch size of 512. Therefore, while larger batch sizes do increase computational load, they do not significantly impede practical application.

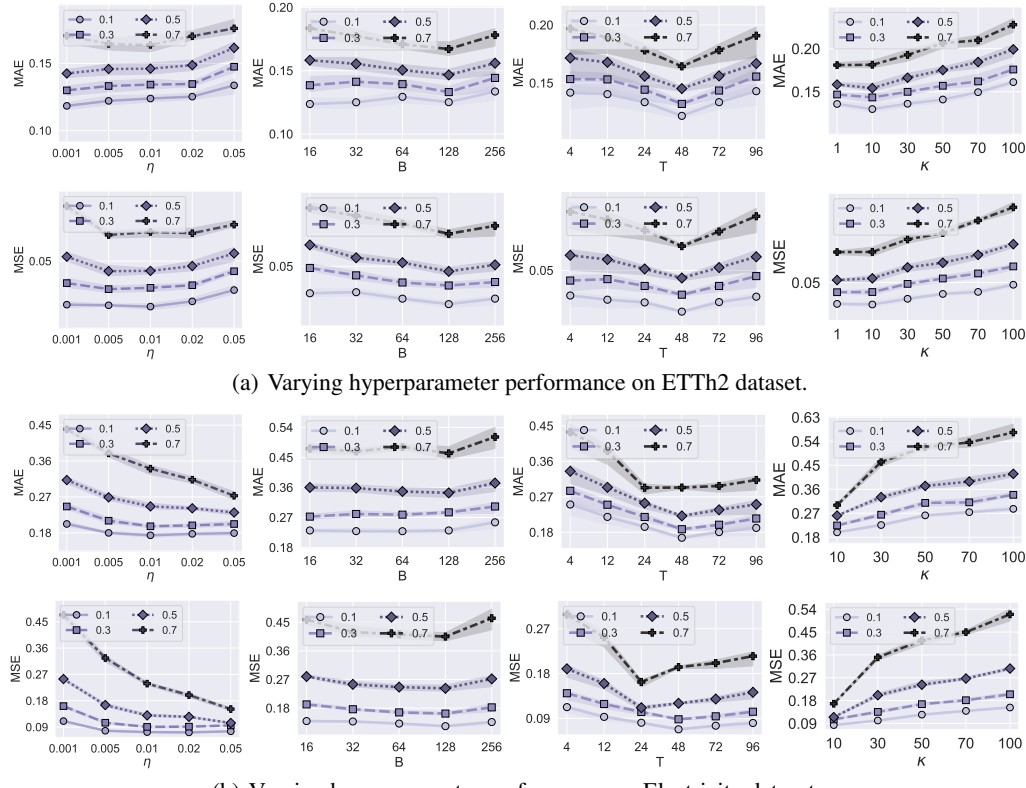

(a) Varying hyperparameter performance on ETTh2 dataset.

(b) Varying hyperparameter performance on Electricity dataset.

Figure 7: Hyperparameter sensitivity results on ETTh2 and Electricity datasets, with missing ratios being 0.1, 0.3, 0.5 and 0.7.

- Holding $B = 128$, we observe a direct, but modest, positive correlation between ($\mathbf{T}$) and running time. Nevertheless, an increase in $T$ does not alter the size of the transport matrix which is primarily influenced by $B$. Therefore, the impact of $T$ on the computation cost is relatively minor.

- An interesting observation is the effect of varying $\kappa$. As $\kappa$ decreases, the running time increases. This trend likely arises from the complexity of achieving a more concrete transport strategy, which requires more iterations to reach a plateau.

## D.2  FULL RESULTS

Table 5 provides a comprehensive comparison of PSW-I's performance against baseline methods, with results for missing ratios of 0.1, 0.3, 0.5, and 0.7 listed separately. Additionally, the standard deviation calculated from five random seeds is presented in Table 6.

## D.3  PARAMETER SENSITIVITY ANALYSIS

We examine the sensitivity of hyperparameters on two additional datasets—Electricity and ETTh2—supplementing the results presented in the main text, as shown in Fig. 7.

## D.4  DOWNSTREAM TASK PERFORMANCE

We evaluate the prediction performance of the iTransformer trained on the datasets imputed using PSW-I and baseline methods. As shown in Table 7, all methods demonstrate comparable performance; however, our PSW-I method achieves the best results, particularly on the ETTh2 dataset. This highlights the effectiveness of our approach in downstream tasks.

Table 5: Full comparison results with missing ratios: 0.1, 0.3, 0.5, 0.7. The length of history window is set to 96 for all baselines. *Avg.* indicates the results averaged over missing ratios.

| Datasets | | ETTh1 | | ETTh2 | | ETTm1 | | ETTm2 | | Electricity | | Traffic | | Weather | | Illness | | Exchange | | PEMS03 | |
|---|---|---|---|---|---|---|---|---|---|---|---|---|---|---|---|---|---|---|---|---|---|
| Metrics | | MSE | MAE | MSE | MAE | MSE | MAE | MSE | MAE | MSE | MAE | MSE | MAE | MSE | MAE | MSE | MAE | MSE | MAE | MSE | MAE |
| Transformer | 0.1 | 0.096 | 0.223 | 0.164 | 0.193 | 0.027 | 0.119 | 0.017 | 0.088 | 0.101 | 0.209 | 0.188 | 0.184 | 0.161 | 0.109 | 0.135 | 0.232 | 0.122 | 0.102 | 0.06 | 0.158 |
| | 0.3 | 0.116 | 0.244 | 0.157 | 0.288 | 0.042 | 0.13 | 0.033 | 0.129 | 0.108 | 0.215 | 0.199 | 0.192 | 0.19 | 0.129 | 0.216 | 0.271 | 0.137 | 0.113 | 0.063 | 0.159 |
| | 0.5 | 0.249 | 0.359 | 0.25 | 0.36 | 0.069 | 0.173 | 0.032 | 0.122 | 0.12 | 0.229 | 0.237 | 0.239 | 0.192 | 0.129 | 0.292 | 0.336 | 0.261 | 0.215 | 0.079 | 0.179 |
| | 0.7 | 0.426 | 0.461 | 0.313 | 0.405 | 0.1 | 0.219 | 0.08 | 0.199 | 0.149 | 0.26 | 0.239 | 0.242 | 0.236 | 0.16 | 0.316 | 0.362 | 0.378 | 0.313 | 0.121 | 0.239 |
| | Avg. | 0.222 | 0.322 | 0.221 | 0.312 | 0.06 | 0.16 | 0.041 | 0.134 | 0.12 | 0.228 | 0.216 | 0.214 | 0.195 | 0.132 | 0.24 | 0.3 | 0.224 | 0.186 | 0.081 | 0.184 |
| DLinear | 0.1 | 0.088 | 0.213 | 0.073 | 0.201 | 0.067 | 0.182 | 0.053 | 0.163 | 0.101 | 0.237 | 0.244 | 0.269 | 0.251 | 0.17 | 0.121 | 0.18 | 0.225 | 0.189 | 0.108 | 0.259 |
| | 0.3 | 0.117 | 0.242 | 0.081 | 0.202 | 0.083 | 0.201 | 0.07 | 0.188 | 0.125 | 0.262 | 0.253 | 0.277 | 0.276 | 0.187 | 0.149 | 0.243 | 0.246 | 0.201 | 0.113 | 0.267 |
| | 0.5 | 0.148 | 0.272 | 0.117 | 0.24 | 0.11 | 0.229 | 0.089 | 0.206 | 0.142 | 0.274 | 0.249 | 0.268 | 0.271 | 0.182 | 0.284 | 0.323 | 0.252 | 0.208 | 0.114 | 0.259 |
| | 0.7 | 0.225 | 0.341 | 0.159 | 0.283 | 0.152 | 0.27 | 0.125 | 0.248 | 0.181 | 0.31 | 0.257 | 0.273 | 0.296 | 0.2 | 0.286 | 0.346 | 0.323 | 0.267 | 0.114 | 0.258 |
| | Avg. | 0.144 | 0.267 | 0.108 | 0.231 | 0.103 | 0.221 | 0.084 | 0.201 | 0.137 | 0.271 | 0.251 | 0.272 | 0.274 | 0.185 | 0.21 | 0.273 | 0.261 | 0.216 | 0.112 | 0.261 |
| TimesNet | 0.1 | 0.173 | 0.291 | 0.084 | 0.214 | 0.032 | 0.127 | 0.031 | 0.135 | 0.1 | 0.226 | 0.159 | 0.217 | 0.251 | 0.17 | 0.084 | 0.185 | 0.238 | 0.2 | 0.057 | 0.168 |
| | 0.3 | 0.154 | 0.291 | 0.154 | 0.284 | 0.043 | 0.147 | 0.053 | 0.17 | 0.114 | 0.239 | 0.175 | 0.225 | 0.252 | 0.17 | 0.31 | 0.311 | 0.296 | 0.243 | 0.06 | 0.167 |
| | 0.5 | 0.163 | 0.297 | 0.126 | 0.26 | 0.061 | 0.179 | 0.072 | 0.197 | 0.124 | 0.249 | 0.205 | 0.246 | 0.272 | 0.183 | 0.278 | 0.304 | 0.343 | 0.283 | 0.077 | 0.194 |
| | 0.7 | 0.523 | 0.532 | 0.168 | 0.295 | 0.11 | 0.238 | 0.114 | 0.243 | 0.18 | 0.302 | 0.266 | 0.283 | 0.344 | 0.233 | 0.252 | 0.306 | 0.401 | 0.332 | 0.109 | 0.23 |
| | Avg. | 0.253 | 0.353 | 0.133 | 0.263 | 0.061 | 0.173 | 0.068 | 0.186 | 0.129 | 0.254 | 0.201 | 0.243 | 0.28 | 0.189 | 0.231 | 0.277 | 0.319 | 0.264 | 0.076 | 0.19 |
| FreTS | 0.1 | 0.127 | 0.256 | 0.18 | 0.271 | 0.053 | 0.16 | 0.041 | 0.143 | 0.135 | 0.27 | 0.23 | 0.275 | 0.168 | 0.114 | 0.092 | 0.186 | 0.233 | 0.196 | 0.126 | 0.283 |
| | 0.3 | 0.145 | 0.282 | 0.091 | 0.211 | 0.043 | 0.141 | 0.034 | 0.126 | 0.133 | 0.263 | 0.229 | 0.27 | 0.169 | 0.114 | 0.196 | 0.294 | 0.221 | 0.181 | 0.097 | 0.239 |
| | 0.5 | 0.229 | 0.358 | 0.141 | 0.261 | 0.048 | 0.149 | 0.037 | 0.134 | 0.168 | 0.292 | 0.236 | 0.27 | 0.174 | 0.117 | 0.295 | 0.349 | 0.256 | 0.211 | 0.116 | 0.259 |
| | 0.7 | 0.236 | 0.351 | 0.176 | 0.294 | 0.074 | 0.187 | 0.042 | 0.137 | 0.185 | 0.315 | 0.241 | 0.267 | 0.2 | 0.135 | 0.532 | 0.472 | 0.201 | 0.166 | 0.097 | 0.227 |
| | Avg. | 0.184 | 0.312 | 0.147 | 0.259 | 0.055 | 0.159 | 0.039 | 0.135 | 0.155 | 0.285 | 0.234 | 0.27 | 0.178 | 0.12 | 0.278 | 0.325 | 0.228 | 0.189 | 0.109 | 0.252 |
| PatchTST | 0.1 | 0.118 | 0.247 | 0.102 | 0.237 | 0.044 | 0.152 | 0.029 | 0.124 | 0.099 | 0.224 | 0.223 | 0.228 | 0.241 | 0.164 | 0.115 | 0.217 | 0.299 | 0.252 | 0.054 | 0.165 |
| | 0.3 | 0.133 | 0.264 | 0.116 | 0.251 | 0.044 | 0.137 | 0.024 | 0.106 | 0.116 | 0.243 | 0.228 | 0.233 | 0.232 | 0.157 | 0.321 | 0.443 | 0.196 | 0.161 | 0.056 | 0.164 |
| | 0.5 | 0.153 | 0.284 | 0.137 | 0.269 | 0.046 | 0.139 | 0.029 | 0.116 | 0.136 | 0.259 | 0.235 | 0.24 | 0.182 | 0.122 | 1.045 | 0.695 | 0.204 | 0.169 | 0.093 | 0.224 |
| | 0.7 | 0.278 | 0.392 | 0.15 | 0.275 | 0.064 | 0.166 | 0.036 | 0.127 | 0.203 | 0.32 | 0.253 | 0.249 | 0.332 | 0.224 | 0.942 | 0.663 | 0.25 | 0.206 | 0.059 | 0.163 |
| | Avg. | 0.171 | 0.297 | 0.126 | 0.258 | 0.05 | 0.149 | 0.03 | 0.118 | 0.138 | 0.262 | 0.235 | 0.238 | 0.247 | 0.167 | 0.605 | 0.505 | 0.237 | 0.197 | 0.065 | 0.179 |
| SCINet | 0.1 | 0.085 | 0.209 | 0.084 | 0.21 | 0.039 | 0.136 | 0.029 | 0.121 | 0.084 | 0.198 | 0.297 | 0.329 | 0.178 | 0.12 | 0.135 | 0.186 | 0.261 | 0.219 | 0.099 | 0.244 |
| | 0.3 | 0.121 | 0.249 | 0.085 | 0.208 | 0.053 | 0.16 | 0.057 | 0.173 | 0.129 | 0.247 | 0.249 | 0.295 | 0.206 | 0.139 | 0.335 | 0.438 | 0.281 | 0.231 | 0.1 | 0.241 |
| | 0.5 | 0.169 | 0.3 | 0.176 | 0.285 | 0.062 | 0.173 | 0.077 | 0.206 | 0.139 | 0.253 | 0.303 | 0.328 | 0.178 | 0.12 | 1.128 | 0.7 | 0.28 | 0.231 | 0.113 | 0.257 |
| | 0.7 | 0.22 | 0.341 | 0.168 | 0.29 | 0.113 | 0.234 | 0.093 | 0.216 | 0.15 | 0.259 | 0.309 | 0.304 | 0.229 | 0.165 | 0.87 | 0.617 | 0.371 | 0.307 | 0.111 | 0.252 |
| | Avg. | 0.149 | 0.275 | 0.128 | 0.248 | 0.067 | 0.176 | 0.064 | 0.179 | 0.125 | 0.239 | 0.29 | 0.314 | 0.198 | 0.136 | 0.617 | 0.473 | 0.298 | 0.247 | 0.106 | 0.249 |
| iTransformer | 0.1 | 0.101 | 0.219 | 0.069 | 0.175 | 0.044 | 0.141 | 0.028 | 0.113 | 0.084 | 0.209 | 0.178 | 0.23 | 0.131 | 0.089 | 0.131 | 0.215 | 0.042 | 0.036 | 0.058 | 0.171 |
| | 0.3 | 0.141 | 0.263 | 0.087 | 0.197 | 0.044 | 0.137 | 0.024 | 0.104 | 0.106 | 0.234 | 0.221 | 0.249 | 0.162 | 0.109 | 0.375 | 0.317 | 0.057 | 0.048 | 0.07 | 0.191 |
| | 0.5 | 0.168 | 0.289 | 0.095 | 0.21 | 0.054 | 0.154 | 0.029 | 0.117 | 0.144 | 0.267 | 0.271 | 0.279 | 0.208 | 0.14 | 0.356 | 0.333 | 0.058 | 0.045 | 0.088 | 0.216 |
| | 0.7 | 0.244 | 0.353 | 0.155 | 0.263 | 0.082 | 0.191 | 0.057 | 0.167 | 0.177 | 0.295 | 0.34 | 0.319 | 0.252 | 0.17 | 0.403 | 0.376 | 0.116 | 0.095 | 0.103 | 0.233 |
| | Avg. | 0.163 | 0.281 | 0.101 | 0.211 | 0.056 | 0.156 | 0.034 | 0.125 | 0.128 | 0.251 | 0.252 | 0.269 | 0.188 | 0.127 | 0.316 | 0.31 | 0.068 | 0.056 | 0.08 | 0.203 |
| SAITS | 0.1 | 0.119 | 0.226 | 0.109 | 0.191 | 0.027 | 0.109 | 0.021 | 0.088 | 0.1 | 0.202 | 0.217 | 0.207 | 0.099 | 0.067 | 0.078 | 0.154 | 1.003 | 0.843 | 0.078 | 0.191 |
| | 0.3 | 0.173 | 0.276 | 0.151 | 0.227 | 0.04 | 0.131 | 0.029 | 0.109 | 0.106 | 0.208 | 0.22 | 0.204 | 0.131 | 0.089 | 0.122 | 0.178 | 1.003 | 0.823 | 0.083 | 0.19 |
| | 0.5 | 0.229 | 0.321 | 0.184 | 0.264 | 0.064 | 0.169 | 0.057 | 0.156 | 0.115 | 0.215 | 0.225 | 0.205 | 0.138 | 0.093 | 0.245 | 0.273 | 1.007 | 0.831 | 0.084 | 0.188 |
| | 0.7 | 0.343 | 0.395 | 0.287 | 0.341 | 0.094 | 0.207 | 0.061 | 0.164 | 0.137 | 0.237 | 0.234 | 0.21 | 0.159 | 0.107 | 0.223 | 0.261 | 1.006 | 0.833 | 0.085 | 0.187 |
| | Avg. | 0.216 | 0.305 | 0.183 | 0.256 | 0.056 | 0.154 | 0.042 | 0.129 | 0.114 | 0.216 | 0.224 | 0.207 | 0.132 | 0.089 | 0.167 | 0.216 | 1.005 | 0.833 | 0.083 | 0.189 |
| CSDI | 0.1 | 0.098 | 0.223 | 0.087 | 0.25 | 0.043 | 0.14 | 0.263 | 0.074 | 0.174 | 0.188 | 0.278 | 0.316 | 0.099 | 0.067 | 0.092 | 0.186 | 0.049 | 0.041 | 0.045 | 0.141 |
| | 0.3 | 0.132 | 0.246 | 0.092 | 0.258 | 0.051 | 0.15 | 0.049 | 0.076 | 0.2 | 0.235 | 0.282 | 0.288 | 0.101 | 0.068 | 0.475 | 0.417 | 0.085 | 0.069 | 0.301 | 0.225 |
| | 0.5 | 0.161 | 0.293 | 0.096 | 0.269 | 0.059 | 0.139 | 0.186 | 0.126 | 0.602 | 0.293 | 0.323 | 0.342 | 0.21 | 0.141 | 0.506 | 0.444 | 0.125 | 0.139 | 0.051 | 0.148 |
| | 0.7 | 0.213 | 0.312 | 0.117 | 0.273 | 0.252 | 0.277 | 0.136 | 0.174 | 0.855 | 0.359 | 0.342 | 0.351 | 0.224 | 0.152 | 0.351 | 0.487 | 0.143 | 0.161 | 0.063 | 0.167 |
| | Avg. | 0.151 | 0.269 | 0.098 | 0.263 | 0.101 | 0.177 | 0.158 | 0.113 | 0.533 | 0.269 | 0.306 | 0.324 | 0.158 | 0.107 | 0.356 | 0.384 | 0.1 | 0.103 | 0.115 | 0.17 |
| Sinkhorn | 0.1 | 0.758 | 0.585 | 0.712 | 0.55 | 0.885 | 0.67 | 0.846 | 0.642 | 0.974 | 0.816 | 0.982 | 0.729 | 0.677 | 0.562 | 0.422 | 0.299 | 0.641 | 0.539 | 0.941 | 0.843 |
| | 0.3 | 0.878 | 0.643 | 0.761 | 0.575 | 0.921 | 0.689 | 0.874 | 0.66 | 0.984 | 0.819 | 1.012 | 0.729 | 0.738 | 0.653 | 0.577 | 0.358 | 0.684 | 0.561 | 0.96 | 0.851 |
| | 0.5 | 0.91 | 0.669 | 0.82 | 0.614 | 0.95 | 0.708 | 0.905 | 0.678 | 0.992 | 0.823 | 1.05 | 0.729 | 0.819 | 0.722 | 0.557 | 0.395 | 0.74 | 0.611 | 0.978 | 0.86 |
| | 0.7 | 0.962 | 0.701 | 0.934 | 0.673 | 0.979 | 0.725 | 0.948 | 0.702 | 0.997 | 0.825 | 1.001 | 0.729 | 0.939 | 0.751 | 0.64 | 0.476 | 0.826 | 0.684 | 0.992 | 0.867 |
| | Avg. | 0.877 | 0.65 | 0.807 | 0.603 | 0.934 | 0.698 | 0.893 | 0.67 | 0.986 | 0.821 | 1.012 | 0.729 | 0.793 | 0.672 | 0.549 | 0.382 | 0.723 | 0.599 | 0.968 | 0.855 |
| TDM | 0.1 | 0.941 | 0.722 | 0.932 | 0.71 | 0.994 | 0.748 | 0.971 | 0.731 | 0.898 | 0.775 | 0.913 | 0.68 | 0.692 | 0.583 | 0.763 | 0.552 | 0.886 | 0.745 | 0.87 | 0.8 |
| | 0.3 | 1.006 | 0.75 | 0.958 | 0.725 | 0.997 | 0.751 | 0.985 | 0.742 | 0.93 | 0.79 | 0.943 | 0.694 | 0.98 | 0.661 | 0.887 | 0.601 | 0.926 | 0.76 | 0.915 | 0.823 |
| | 0.5 | 0.993 | 0.749 | 0.978 | 0.738 | 1.002 | 0.754 | 0.995 | 0.746 | 0.954 | 0.801 | 0.961 | 0.704 | 0.823 | 0.712 | 0.862 | 0.622 | 0.957 | 0.79 | 0.946 | 0.838 |
| | 0.7 | 0.989 | 0.749 | 1.017 | 0.751 | 1.004 | 0.757 | 1.001 | 0.749 | 0.975 | 0.812 | 0.976 | 0.713 | 0.965 | 0.798 | 0.852 | 0.651 | 0.981 | 0.812 | 0.969 | 0.851 |
| | Avg. | 0.982 | 0.742 | 0.971 | 0.731 | 0.999 | 0.752 | 0.988 | 0.742 | 0.939 | 0.795 | 0.948 | 0.698 | 0.865 | 0.689 | 0.841 | 0.607 | 0.938 | 0.776 | 0.925 | 0.828 |
| PWS-I(Ours) | 0.1 | 0.079 | 0.188 | 0.035 | 0.124 | 0.034 | 0.112 | 0.016 | 0.083 | 0.073 | 0.177 | 0.152 | 0.175 | 0.092 | 0.062 | 0.029 | 0.086 | 0.026 | 0.022 | 0.044 | 0.142 |
| | 0.3 | 0.105 | 0.213 | 0.041 | 0.133 | 0.04 | 0.12 | 0.018 | 0.088 | 0.089 | 0.196 | 0.173 | 0.186 | 0.098 | 0.066 | 0.056 | 0.109 | 0.028 | 0.023 | 0.046 | 0.146 |
| | 0.5 | 0.125 | 0.234 | 0.047 | 0.145 | 0.048 | 0.133 | 0.021 | 0.096 | 0.115 | 0.22 | 0.205 | 0.204 | 0.107 | 0.072 | 0.07 | 0.124 | 0.032 | 0.026 | 0.049 | 0.15 |
| | 0.7 | 0.194 | 0.289 | 0.059 | 0.165 | 0.066 | 0.158 | 0.029 | 0.11 | 0.148 | 0.239 | 0.26 | 0.237 | 0.131 | 0.088 | 0.114 | 0.17 | 0.039 | 0.033 | 0.056 | 0.159 |
| | Avg. | 0.126 | 0.231 | 0.046 | 0.142 | 0.047 | 0.131 | 0.021 | 0.094 | 0.106 | 0.208 | 0.197 | 0.199 | 0.107 | 0.072 | 0.067 | 0.122 | 0.031 | 0.026 | 0.049 | 0.149 |

Table 6: The standard deviation of Table 5.

| Datasets | | ETTh1 | | ETTh2 | | ETTm1 | | ETTm2 | | Electricity | | Traffic | | Weather | | Illness | | Exchange | | PEMS03 | |
|---|---|---|---|---|---|---|---|---|---|---|---|---|---|---|---|---|---|---|---|---|---|
| Metrics | | MSE | MAE | MSE | MAE | MSE | MAE | MSE | MAE | MSE | MAE | MSE | MAE | MSE | MAE | MSE | MAE | MSE | MAE | MSE | MAE |
| Transformer | 0.1 | 0.0 | 0.004 | 0.008 | 0.008 | 0.009 | 0.005 | 0.007 | 0.005 | 0.004 | 0.005 | 0.009 | 0.009 | 0.01 | 0.008 | 0.0 | 0.004 | 0.004 | 0.004 | 0.002 | 0.001 |
| | 0.3 | 0.003 | 0.005 | 0.002 | 0.003 | 0.003 | 0.008 | 0.007 | 0.005 | 0.003 | 0.001 | 0.008 | 0.004 | 0.003 | 0.008 | 0.006 | 0.008 | 0.0 | 0.002 | 0.001 | 0.0 |
| | 0.5 | 0.007 | 0.005 | 0.007 | 0.008 | 0.009 | 0.005 | 0.008 | 0.008 | 0.006 | 0.003 | 0.0 | 0.009 | 0.009 | 0.003 | 0.005 | 0.007 | 0.009 | 0.003 | 0.007 | 0.003 |
| | 0.7 | 0.005 | 0.002 | 0.004 | 0.0 | 0.007 | 0.008 | 0.002 | 0.007 | 0.002 | 0.0 | 0.001 | 0.002 | 0.007 | 0.007 | 0.006 | 0.001 | 0.006 | 0.0 | 0.003 | 0.002 |
| | Avg. | 0.004 | 0.004 | 0.005 | 0.005 | 0.007 | 0.007 | 0.006 | 0.006 | 0.004 | 0.002 | 0.005 | 0.006 | 0.005 | 0.006 | 0.004 | 0.005 | 0.004 | 0.002 | 0.003 | 0.002 |
| DLinear | 0.1 | 0.005 | 0.007 | 0.002 | 0.008 | 0.01 | 0.003 | 0.0 | 0.007 | 0.009 | 0.007 | 0.001 | 0.008 | 0.006 | 0.001 | 0.002 | 0.003 | 0.001 | 0.008 | 0.004 | 0.003 |
| | 0.3 | 0.006 | 0.005 | 0.008 | 0.0 | 0.008 | 0.004 | 0.001 | 0.007 | 0.003 | 0.009 | 0.01 | 0.002 | 0.007 | 0.006 | 0.005 | 0.003 | 0.007 | 0.006 | 0.005 | 0.003 |
| | 0.5 | 0.008 | 0.003 | 0.004 | 0.007 | 0.01 | 0.001 | 0.005 | 0.002 | 0.004 | 0.003 | 0.004 | 0.004 | 0.009 | 0.008 | 0.001 | 0.005 | 0.01 | 0.003 | 0.008 | 0.002 |
| | 0.7 | 0.009 | 0.001 | 0.002 | 0.008 | 0.007 | 0.007 | 0.007 | 0.003 | 0.01 | 0.009 | 0.007 | 0.004 | 0.004 | 0.001 | 0.008 | 0.005 | 0.007 | 0.006 | 0.007 | 0.008 |
| | Avg. | 0.007 | 0.004 | 0.004 | 0.006 | 0.009 | 0.004 | 0.003 | 0.005 | 0.007 | 0.007 | 0.005 | 0.005 | 0.005 | 0.004 | 0.004 | 0.004 | 0.003 | 0.006 | 0.006 | 0.004 |
| TimesNet | 0.1 | 0.008 | 0.006 | 0.009 | 0.006 | 0.004 | 0.009 | 0.005 | 0.006 | 0.001 | 0.009 | 0.004 | 0.01 | 0.005 | 0.002 | 0.003 | 0.007 | 0.009 | 0.006 | 0.003 | 0.009 |
| | 0.3 | 0.004 | 0.001 | 0.007 | 0.01 | 0.005 | 0.007 | 0.004 | 0.001 | 0.007 | 0.006 | 0.006 | 0.002 | 0.01 | 0.005 | 0.009 | 0.007 | 0.002 | 0.001 | 0.004 | 0.002 |
| | 0.5 | 0.008 | 0.005 | 0.005 | 0.0 | 0.002 | 0.0 | 0.001 | 0.01 | 0.003 | 0.004 | 0.007 | 0.002 | 0.003 | 0.001 | 0.005 | 0.006 | 0.008 | 0.007 | 0.005 | 0.007 |
| | 0.7 | 0.007 | 0.01 | 0.002 | 0.005 | 0.005 | 0.002 | 0.002 | 0.005 | 0.002 | 0.006 | 0.002 | 0.001 | 0.004 | 0.002 | 0.01 | 0.003 | 0.009 | 0.004 | 0.001 | 0.004 |
| | Avg. | 0.007 | 0.005 | 0.006 | 0.005 | 0.004 | 0.005 | 0.003 | 0.006 | 0.003 | 0.006 | 0.005 | 0.004 | 0.003 | 0.003 | 0.007 | 0.006 | 0.007 | 0.005 | 0.003 | 0.005 |
| FreTS | 0.1 | 0.007 | 0.003 | 0.004 | 0.007 | 0.005 | 0.005 | 0.002 | 0.009 | 0.005 | 0.005 | 0.002 | 0.005 | 0.001 | 0.01 | 0.001 | 0.002 | 0.008 | 0.006 | 0.002 | 0.003 |
| | 0.3 | 0.002 | 0.008 | 0.007 | 0.003 | 0.005 | 0.009 | 0.004 | 0.005 | 0.0 | 0.01 | 0.001 | 0.008 | 0.007 | 0.008 | 0.003 | 0.001 | 0.007 | 0.001 | 0.001 | 0.003 |
| | 0.5 | 0.008 | 0.003 | 0.004 | 0.001 | 0.001 | 0.0 | 0.007 | 0.005 | 0.003 | 0.008 | 0.001 | 0.008 | 0.001 | 0.003 | 0.007 | 0.0 | 0.002 | 0.002 | 0.009 | 0.005 |
| | 0.7 | 0.003 | 0.006 | 0.006 | 0.009 | 0.008 | 0.007 | 0.008 | 0.002 | 0.001 | 0.008 | 0.005 | 0.007 | 0.008 | 0.006 | 0.0 | 0.006 | 0.002 | 0.009 | 0.009 | 0.007 |
| | Avg. | 0.005 | 0.005 | 0.005 | 0.005 | 0.005 | 0.005 | 0.005 | 0.005 | 0.002 | 0.008 | 0.002 | 0.007 | 0.007 | 0.007 | 0.003 | 0.002 | 0.004 | 0.005 | 0.005 | 0.004 |
| PatchTST | 0.1 | 0.01 | 0.004 | 0.006 | 0.01 | 0.01 | 0.0 | 0.006 | 0.006 | 0.002 | 0.009 | 0.0 | 0.009 | 0.009 | 0.002 | 0.004 | 0.002 | 0.01 | 0.006 | 0.009 | 0.004 |
| | 0.3 | 0.008 | 0.006 | 0.001 | 0.007 | 0.008 | 0.007 | 0.0 | 0.001 | 0.006 | 0.008 | 0.001 | 0.001 | 0.0 | 0.0 | 0.01 | 0.006 | 0.003 | 0.004 | 0.005 | 0.009 |
| | 0.5 | 0.003 | 0.008 | 0.003 | 0.003 | 0.006 | 0.004 | 0.006 | 0.001 | 0.001 | 0.001 | 0.004 | 0.007 | 0.001 | 0.005 | 0.002 | 0.006 | 0.009 | 0.007 | 0.008 | 0.003 |
| | 0.7 | 0.001 | 0.004 | 0.005 | 0.008 | 0.009 | 0.001 | 0.005 | 0.002 | 0.007 | 0.001 | 0.005 | 0.003 | 0.004 | 0.003 | 0.001 | 0.004 | 0.005 | 0.009 | 0.002 | 0.005 |
| | Avg. | 0.006 | 0.006 | 0.004 | 0.007 | 0.008 | 0.003 | 0.004 | 0.003 | 0.004 | 0.005 | 0.003 | 0.005 | 0.005 | 0.003 | 0.004 | 0.005 | 0.002 | 0.007 | 0.006 | 0.005 |
| SCINet | 0.1 | 0.004 | 0.0 | 0.009 | 0.004 | 0.002 | 0.0 | 0.009 | 0.01 | 0.005 | 0.007 | 0.0 | 0.004 | 0.007 | 0.005 | 0.008 | 0.006 | 0.001 | 0.004 | 0.003 | 0.003 |
| | 0.3 | 0.0 | 0.002 | 0.008 | 0.007 | 0.009 | 0.006 | 0.006 | 0.0 | 0.0 | 0.002 | 0.003 | 0.006 | 0.01 | 0.002 | 0.006 | 0.002 | 0.008 | 0.007 | 0.006 | 0.005 |
| | 0.5 | 0.001 | 0.005 | 0.004 | 0.008 | 0.001 | 0.004 | 0.001 | 0.0 | 0.006 | 0.008 | 0.006 | 0.001 | 0.009 | 0.001 | 0.005 | 0.008 | 0.001 | 0.003 | 0.004 | 0.007 |
| | 0.7 | 0.001 | 0.006 | 0.001 | 0.007 | 0.005 | 0.009 | 0.007 | 0.004 | 0.001 | 0.001 | 0.002 | 0.009 | 0.0 | 0.005 | 0.003 | 0.006 | 0.007 | 0.006 | 0.01 | 0.009 |
| | Avg. | 0.002 | 0.003 | 0.006 | 0.006 | 0.004 | 0.005 | 0.006 | 0.004 | 0.003 | 0.005 | 0.003 | 0.005 | 0.004 | 0.003 | 0.005 | 0.005 | 0.004 | 0.005 | 0.006 | 0.006 |
| iTransformer | 0.1 | 0.004 | 0.008 | 0.005 | 0.009 | 0.001 | 0.008 | 0.005 | 0.001 | 0.006 | 0.0 | 0.007 | 0.001 | 0.004 | 0.0 | 0.004 | 0.004 | 0.006 | 0.003 | 0.008 | 0.002 |
| | 0.3 | 0.005 | 0.008 | 0.006 | 0.001 | 0.005 | 0.006 | 0.008 | 0.005 | 0.007 | 0.009 | 0.003 | 0.001 | 0.008 | 0.005 | 0.01 | 0.003 | 0.007 | 0.006 | 0.009 | 0.003 |
| | 0.5 | 0.007 | 0.009 | 0.004 | 0.004 | 0.003 | 0.002 | 0.009 | 0.01 | 0.004 | 0.007 | 0.009 | 0.006 | 0.005 | 0.002 | 0.003 | 0.001 | 0.002 | 0.007 | 0.004 | 0.005 |
| | 0.7 | 0.004 | 0.007 | 0.007 | 0.003 | 0.003 | 0.008 | 0.001 | 0.004 | 0.007 | 0.01 | 0.005 | 0.0 | 0.003 | 0.003 | 0.004 | 0.006 | 0.008 | 0.005 | 0.01 | 0.005 |
| | Avg. | 0.005 | 0.008 | 0.005 | 0.004 | 0.003 | 0.006 | 0.006 | 0.005 | 0.006 | 0.007 | 0.006 | 0.002 | 0.005 | 0.003 | 0.005 | 0.004 | 0.006 | 0.005 | 0.008 | 0.004 |
| SAITS | 0.1 | 0.007 | 0.008 | 0.008 | 0.007 | 0.008 | 0.007 | 0.007 | 0.001 | 0.003 | 0.007 | 0.001 | 0.006 | 0.003 | 0.008 | 0.002 | 0.004 | 0.008 | 0.009 | 0.009 | 0.001 |
| | 0.3 | 0.007 | 0.009 | 0.005 | 0.009 | 0.007 | 0.006 | 0.002 | 0.007 | 0.01 | 0.006 | 0.004 | 0.006 | 0.005 | 0.003 | 0.007 | 0.007 | 0.007 | 0.006 | 0.008 | 0.005 |
| | 0.5 | 0.009 | 0.008 | 0.004 | 0.01 | 0.001 | 0.008 | 0.001 | 0.008 | 0.009 | 0.006 | 0.005 | 0.001 | 0.009 | 0.009 | 0.004 | 0.005 | 0.004 | 0.004 | 0.002 | 0.001 |
| | 0.7 | 0.005 | 0.004 | 0.001 | 0.004 | 0.002 | 0.002 | 0.009 | 0.009 | 0.001 | 0.003 | 0.001 | 0.009 | 0.001 | 0.007 | 0.003 | 0.002 | 0.01 | 0.0 | 0.005 | 0.007 |
| | Avg. | 0.007 | 0.007 | 0.005 | 0.008 | 0.005 | 0.006 | 0.005 | 0.006 | 0.006 | 0.006 | 0.003 | 0.005 | 0.004 | 0.007 | 0.004 | 0.005 | 0.005 | 0.005 | 0.006 | 0.004 |
| CSDI | 0.1 | 0.003 | 0.005 | 0.009 | 0.003 | 0.0 | 0.005 | 0.001 | 0.001 | 0.004 | 0.001 | 0.006 | 0.003 | 0.005 | 0.008 | 0.005 | 0.01 | 0.008 | 0.004 | 0.005 | 0.005 |
| | 0.3 | 0.002 | 0.01 | 0.005 | 0.007 | 0.008 | 0.002 | 0.007 | 0.003 | 0.001 | 0.0 | 0.003 | 0.004 | 0.009 | 0.004 | 0.002 | 0.004 | 0.002 | 0.009 | 0.001 | 0.009 |
| | 0.5 | 0.005 | 0.001 | 0.001 | 0.0 | 0.009 | 0.009 | 0.002 | 0.008 | 0.001 | 0.003 | 0.005 | 0.008 | 0.002 | 0.004 | 0.004 | 0.006 | 0.0 | 0.008 | 0.002 | 0.003 |
| | 0.7 | 0.002 | 0.01 | 0.004 | 0.002 | 0.004 | 0.006 | 0.002 | 0.005 | 0.006 | 0.001 | 0.006 | 0.004 | 0.009 | 0.002 | 0.008 | 0.009 | 0.005 | 0.01 | 0.001 | 0.008 |
| | Avg. | 0.003 | 0.007 | 0.005 | 0.003 | 0.005 | 0.005 | 0.003 | 0.004 | 0.003 | 0.001 | 0.005 | 0.005 | 0.005 | 0.005 | 0.005 | 0.007 | 0.004 | 0.008 | 0.002 | 0.006 |
| Sinkhorn | 0.1 | 0.002 | 0.007 | 0.001 | 0.005 | 0.002 | 0.006 | 0.004 | 0.003 | 0.003 | 0.006 | 0.009 | 0.001 | 0.001 | 0.006 | 0.01 | 0.002 | 0.009 | 0.008 | 0.01 | 0.002 |
| | 0.3 | 0.001 | 0.008 | 0.009 | 0.003 | 0.002 | 0.002 | 0.01 | 0.006 | 0.002 | 0.009 | 0.002 | 0.001 | 0.006 | 0.01 | 0.0 | 0.01 | 0.003 | 0.001 | 0.009 | 0.001 |
| | 0.5 | 0.008 | 0.007 | 0.002 | 0.01 | 0.005 | 0.003 | 0.001 | 0.008 | 0.009 | 0.002 | 0.005 | 0.002 | 0.008 | 0.008 | 0.008 | 0.005 | 0.005 | 0.001 | 0.005 | 0.004 |
| | 0.7 | 0.009 | 0.004 | 0.008 | 0.001 | 0.005 | 0.01 | 0.002 | 0.003 | 0.0 | 0.001 | 0.002 | 0.009 | 0.009 | 0.008 | 0.004 | 0.003 | 0.002 | 0.005 | 0.005 | 0.001 |
| | Avg. | 0.005 | 0.006 | 0.005 | 0.005 | 0.004 | 0.005 | 0.004 | 0.005 | 0.003 | 0.005 | 0.005 | 0.003 | 0.005 | 0.008 | 0.006 | 0.005 | 0.007 | 0.004 | 0.007 | 0.002 |
| TDM | 0.1 | 0.003 | 0.006 | 0.007 | 0.009 | 0.0 | 0.007 | 0.004 | 0.003 | 0.006 | 0.001 | 0.002 | 0.002 | 0.008 | 0.001 | 0.002 | 0.007 | 0.004 | 0.001 | 0.005 | 0.002 |
| | 0.3 | 0.0 | 0.0 | 0.004 | 0.004 | 0.005 | 0.005 | 0.006 | 0.005 | 0.01 | 0.009 | 0.003 | 0.004 | 0.0 | 0.0 | 0.009 | 0.009 | 0.008 | 0.001 | 0.006 | 0.002 |
| | 0.5 | 0.004 | 0.008 | 0.007 | 0.009 | 0.003 | 0.005 | 0.0 | 0.004 | 0.005 | 0.007 | 0.009 | 0.004 | 0.002 | 0.006 | 0.007 | 0.007 | 0.005 | 0.006 | 0.003 | 0.007 |
| | 0.7 | 0.008 | 0.008 | 0.007 | 0.007 | 0.003 | 0.006 | 0.008 | 0.0 | 0.01 | 0.001 | 0.002 | 0.006 | 0.004 | 0.008 | 0.007 | 0.004 | 0.001 | 0.009 | 0.0 | 0.001 |
| | Avg. | 0.004 | 0.005 | 0.006 | 0.007 | 0.003 | 0.006 | 0.005 | 0.003 | 0.008 | 0.004 | 0.004 | 0.004 | 0.003 | 0.004 | 0.006 | 0.007 | 0.003 | 0.004 | 0.003 | 0.003 |
| PSW-I(Ours) | 0.1 | 0.008 | 0.004 | 0.007 | 0.002 | 0.001 | 0.002 | 0.003 | 0.003 | 0.008 | 0.005 | 0.007 | 0.005 | 0.005 | 0.01 | 0.005 | 0.005 | 0.0 | 0.002 | 0.008 | 0.009 |
| | 0.3 | 0.004 | 0.008 | 0.005 | 0.002 | 0.009 | 0.009 | 0.004 | 0.009 | 0.002 | 0.001 | 0.002 | 0.004 | 0.004 | 0.003 | 0.007 | 0.004 | 0.001 | 0.005 | 0.009 | 0.0 |
| | 0.5 | 0.003 | 0.009 | 0.006 | 0.003 | 0.0 | 0.008 | 0.008 | 0.007 | 0.003 | 0.002 | 0.002 | 0.003 | 0.0 | 0.009 | 0.006 | 0.007 | 0.005 | 0.005 | 0.003 | 0.006 |
| | 0.7 | 0.004 | 0.006 | 0.001 | 0.003 | 0.003 | 0.009 | 0.006 | 0.002 | 0.002 | 0.009 | 0.002 | 0.005 | 0.002 | 0.001 | 0.007 | 0.006 | 0.009 | 0.004 | 0.009 | 0.007 |
| | Avg. | 0.005 | 0.007 | 0.005 | 0.003 | 0.003 | 0.007 | 0.005 | 0.005 | 0.004 | 0.004 | 0.003 | 0.004 | 0.004 | 0.006 | 0.006 | 0.005 | 0.006 | 0.004 | 0.007 | 0.005 |

Table 7: Downstream task performance with iTransformer as the prediction model.

| Datasets | ETTh1 | | ETTh2 | | Exchange | | Weather | |
|---|---|---|---|---|---|---|---|---|
| Metrics | MSE | MAE | MSE | MAE | MSE | MAE | MSE | MAE |
| Transformer | 0.389 | 0.406 | 0.299 | 0.385 | 0.093 | 0.215 | 0.177 | 0.218 |
| DLinear | 0.389 | 0.406 | 0.299 | 0.386 | 0.091 | 0.212 | 0.178 | 0.219 |
| TimesNet | 0.390 | 0.407 | 0.302 | 0.388 | 0.098 | 0.222 | 0.179 | 0.218 |
| SCINet | 0.389 | 0.406 | 0.296 | 0.383 | 0.098 | 0.222 | 0.179 | 0.219 |
| iTransformer | 0.389 | 0.406 | 0.293 | 0.381 | 0.093 | 0.215 | 0.179 | 0.219 |
| SAITS | 0.389 | 0.406 | 0.293 | 0.381 | 0.097 | 0.220 | 0.176 | 0.216 |
| CSDI | 0.389 | 0.405 | 0.297 | 0.384 | 0.098 | 0.222 | 0.176 | 0.216 |
| **PSW-I(Ours)** | 0.387 | 0.401 | 0.279 | 0.371 | 0.091 | 0.212 | 0.175 | 0.215 |

