# OpenReview forum: "Optimal Transport for Time Series Imputation"
_ICLR.cc/2025/Conference — ICLR 2025 Poster_

### Official Review · Reviewer_4Wtk · 2024-11-02

**Soundness:** 2
**Presentation:** 2
**Contribution:** 2
**Rating:** 5
**Confidence:** 2

**Summary:**

The paper proposes an optimal transport (OT) based time-series imputation method. The authors claim that naive application of OT does not work for time-series data. The proposed method consider applying OT in the frequency domain of the original data, called pairwise spectrum distance (PSD). Further, to deal with multiple modes, proximal spectral Wasserstein (PSW) distance is also proposed, in which mass constraint is removed to make transportation more flexible.

**Strengths:**

- Using OT to impute time-series data is an interesting approach.

- Empirical evaluation shows high performance.

**Weaknesses:**

- Some technical justification is vague. Clearer descriptions would be desired.

- Introduction is a bit too abstract about the proposed method. It describes what problem is solved in the paper, but does not describe the basic technical idea how it is achieved.

**Questions:**

- In 'Contributions', the authors mention that PSW-I eliminates the need for masking, but PSW-I seems to use masking (eg as shown in Fig3). What does this description mean?

- What does Lemma 3.2 indicate? How do you know 'deviates more from the typical elements of \beta'? Further, how do you know the PSW discrepancy avoid the problem indicated by this lemma? In Theorem C.2, the perturbation in PSW discrepancy is shown. Is it possible to compare this with Lemma 3.2? Even if it is possible, what does it mean? Is it clear how large (or small) perturbation caused by outliers ultimately affect the final imputation?

- Although D_KL is used in (2), T 1_m and T^T 1_n is not a probability distribution (because the normalization (sum equals 1) constraint is removed). How are they normalized?

- What is the definition of the 'DFT matrix' in the gradient? How does the gradient of (2) become \Delta_alpha_i P and \Delta_beta_j P? The second term in (2) disappear? The optimality condition wrt T is considered in this gradient calculation?

- After (1), D is squared distance, while in definition 3.1 D is distance without square. Which is correct?

Minor issues:

- In the end of p3: Fig. 4 should be Fig 1? In Fig 1(a), the left is W^(F)?

- The first word of Sec3.2: 'time-series' -> 'Time-series'

---

> ### Author Response · Authors · 2024-11-20
>
> **Thank you very much for your meticulous comments and appreciation of our novelty and empirical results. We have noticed that this reviewer has carefully checked the formulas and theorems in both main text and appendix and raises a bag of constructive suggestions. Below are our responses to the specific concerns and queries.**
>
> ---
>
> #### **[W1] Some technical justification is vague. Clearer descriptions would be desired.**
> **Response.**  It seems that this concern manifests as the specific comments in the `question` section. We sincerely appreciate these meticulous comments, and have made careful revisions to improve the clarity of technical justification. Please kindly see our response to these questions below.
>
> #### **[W2] Introduction is a bit too abstract about the proposed method. It describes what problem is solved in the paper, but does not describe the basic technical idea how it is achieved.**
> **Response.** Thank you very much for your sincere comment. We have made extensive editing works on the 5-th paragraph of introduction, with particular focus on highlighting the basic technical idea of PSW-I to tackle the research problem. The revised paragraph is attached below.
> > To this end, we propose the Proximal Spectrum Wasserstein (PSW) discrepancy, a novel discrepancy tailored for comparing sets of time-series based on optimal transport. Specifically, PSW integrates a pairwise spectral distance, which transforms time-series into the frequency domain and then calculate the pair-wise absolute difference. By comparing time series in the frequency domain, the underlying temporal dependencies and patterns are captured. Moreover, PSW incorporates selective matching regularization, which relaxes the hard matching constraints of traditional optimal transport and allows for flexible mass matching between distributions. This relaxation enhances robustness to non-stationary. Building upon PSW, we develop the PSW for Imputation (PSW-I) framework, which iteratively refines the imputed missing values by minimizing the PSW discrepancy. Extensive experiments demonstrate that PSW-I effectively captures temporal patterns and accommodates non-stationary behaviors, significantly outperforming existing time-series imputation methods.
>
> #### **[Q1] In 'Contributions', the authors mention that PSW-I eliminates the need for masking, but PSW-I seems to use masking (eg as shown in Fig3). What does this description mean?**
> **Response.** Thank you very much for your query. In our manuscript, the mask matrix $\mathbf{M}$ is utilized exclusively to indicate the missingness of data at each indice. The non-missing indices are employed as the training\&validation dataset and the missing indices are used for test. In contrast, many prevailing works such as CSDI and time-series foundational models (iTransformer, TimesNet, etc) perform **an additional masking operation** on the training dataset to generate labels for model training. Our method bypasses the complexity of this additional masking operation, which proves to be beneficial for imputation quality [1].

---

> ### Author Response · Authors · 2024-11-20
>
> #### **[Q2] What does Lemma 3.2 indicate? How do you know 'deviates more from the typical elements of \beta'? Further, how do you know the PSW discrepancy avoid the problem indicated by this lemma? In Theorem C.2, the perturbation in PSW discrepancy is shown. Is it possible to compare this with Lemma 3.2? Even if it is possible, what does it mean? Is it clear how large (or small) perturbation caused by outliers ultimately affect the final imputation?**
> **Response.** Thank you for your invaluable suggestion for a clearer exposition of Lemma 3.2 and its relation to Theorem C.2. To this end, we have revised Theorem C.2 to make it comparable to Lemma 3.2.
> - **Firstly, we delineate Lemma 3.2 and Theorem C.2.** Suppose $\tilde{\alpha}=\zeta\delta_{z}+(1-\zeta)\alpha$ is a distribution perturbed by a Dirac mode at $z$ with relative mass $\zeta \in (0,1)$, $\mathcal{W}$ is the standard wasserstein discrepancy, $\mathcal{P}^\kappa$ is the PSW with matching strength $\kappa$. **Lemma 3.2** states that: for an arbitrary sample $y_*$ in $\beta$:
>     $$\mathcal{W}(\tilde{\alpha}, \beta) \geq(1-\zeta) \mathcal{W}(\alpha, \beta)+\zeta\left(D\left(z, y^*\right)-\mathrm{const}\right)$$
>     where $D\left(z, y^*\right)$ is the deviation of $\delta_z$, $\mathrm{const}$ encapsulates the terms irrelative to $\zeta$ and $D$. In contrast, **Theorem C.2** states that:
>     $$\mathcal{P}^\kappa(\tilde{\alpha}, \beta)\leq (1-\zeta) \mathcal{P}^\kappa(\alpha,\beta) + 2\kappa\zeta(1-e^{-d(z)/2\kappa}).$$
>     where $d(z)$ is the average distance between $z$ and samples in $\beta$.
>
> - **Secondly, we clarify and COMPARE the implications of Lemma 3.2 and Theorem C.2.**
>   - Lemma 3.2 demonstrates that the standard Wasserstein distance  $\mathcal{W}(\tilde{\alpha}, \beta)$ increases proportionally to the deviation  $D(z, y^*)$  of the outlier from the typical elements $y^*$ of  $\beta$. $\mathcal{W}(\tilde{\alpha}, \beta)$ **grows without bound** as the deviation increases, indicating sensitivity to outliers.
>   - Theorem C.2 demonstrates that $\mathcal{P}^\kappa(\tilde{\alpha}, \beta)$ **remains bounded** even if the outlier’s deviation $d(z)\rightarrow \infty$. What remains is the PSW excluding the outlier mode and a cost bounded by $2\kappa(1-\zeta)$.
>   - **By comparing Lemma 3.2 and Theorem C.2, it becomes evident that the standard Wasserstein distance is susceptible to large perturbations from outliers, but the PSW discrepancy mitigates this issue by capping the impact of such deviations. Consequently, PSW provides a more stable and reliable discrepancy amidst outlier modes.**
> - Finally, we note that in our framework, accurate imputation relies on precise discrepancy estimation. False discrepancy estimation  produces false gradient and thereby misleading the update of imputation. In this context, the robustness of PSW, demonstrated in Theorem C.2, matters since it offers reliable discrepancy estimation and subsequently gradient estimation amidst outlier modes, which enhances the accuracy of imputation.
>
> #### **[Q3] Although D_KL is used in (2), $T 1_m$ and $T^T 1_n$ is not a probability distribution (because the normalization (sum equals 1) constraint is removed). How are they normalized?**
> **Response.** In this work, the KL divergence, for instance $\mathrm{D}\_\mathrm{KL}(T 1_m,\Delta_n)$, is employed merely as a tool to align the vectors $T 1_m$ and $\Delta_n$. Specifically, denote $a=T 1_m$ and $b=\Delta_n$, for the $i$-th element, if $a_i>b_i$, then $\mathrm{D}\_\mathrm{KL}(a_i,b_i)>0$, where the optimization objective encourages to reduce $a_i$ towards $b_i$. Otherwise, $\mathrm{D}\_\mathrm{KL}(a,b)<0$, where the optimization objective encourages to increase $a_i$ towards $b_i$. **Therefore, we observe that KL divergence effectively guides the alignment of $a$ and $b$ irrespective of whether their sums equal to 1.**

---

> ### Author Response · Authors · 2024-11-20
>
> #### **[Q4] What is the definition of the 'DFT matrix' in the gradient? How does the gradient of (2) become $\nabla_{\alpha_i} P$ and $\nabla_{\beta_j} P$? The second term in (2) disappear? The optimality condition wrt T is considered in this gradient calculation?**
> **Response.** Thank you very much for your meticulous and valuable comment. Below we answer this question by (1) clarifying the definition of the DFT matrix and (2) delineating the derivation of $\nabla_{\alpha_i} P$ and $\nabla_{\beta_j} P$.
> - To clarify the definition of the DFT matrix, we recall that for a sequence $\mathbf{x}=[\mathbf{x}\_0, ..., \mathbf{x}\_\mathrm{T-1}]$, the DFT is calculated as $$\mathbf{x}\^\mathrm{(F)}\_k=\sum\_{t=0}\^\mathrm{\mathrm{T}-1} \mathbf{x}\_t e^{-j2\pi k t/\mathrm{T}}=[\mathbf{x}\_0,\mathbf{x}\_1,...,\mathbf{x}\_\mathrm{\mathrm{T}-1}]\cdot [e\^{-j2\pi k 0/\mathrm{T}},e\^{-j2\pi k 1/\mathrm{T}},...,e\^{-j2\pi k (\mathrm{T-1})/\mathrm{T}}]^\top,$$ where $\mathbf{x}\^\mathrm{(F)}\_k$ is the $k$-th element of the spectrum $\mathbf{x}\^\mathrm{(F)}$. On the basis, denote $\mathbf{W}\^\mathrm{(F)}=[\mathbf{W}\^\mathrm{(F)}\_{t,k}]\_{t,k=0}\^\mathrm{T-1}$ where $\mathbf{W}^\mathrm{(F)}_{t,k}=e^{-j2\pi k t/\mathrm{T}}$ is the t-th row and k-th column of $\mathbf{W}^\mathrm{(F)}$, the spectrum generated by DFT can be represented as $$\mathbf{x}^\mathrm{(F)}=\mathbf{x}\mathbf{W}^\mathrm{(F)},$$ which clarifies the definition and role of the DFT matrix $\mathbf{W}^\mathrm{(F)}$.
>   - **Action.** We have added Definition C.5 to delineate the DFT process where the role of the DFT matrix $\mathbf{W}^\mathrm{(F)}$ is clarified.
>
> - After acquiring the optimum transport matrix by solving the optimization problem in Eq.(2), denoted as $\mathbf{T}$, the PSW is calculated as
> $$\mathcal{P}(\alpha, \beta) := \left\langle \mathbf{D}\^\mathrm{(F)}, \mathbf{T} \right\rangle + \kappa \left( \mathrm{D\_{KL}}(\mathbf{T} \mathbf{1}\_m \Vert \Delta\_n) + \mathrm{D\_{KL}}(\mathbf{T}\^\mathrm{T} \mathbf{1}\_n \Vert \Delta_m) \right)$$
> which depends on $\alpha$ and $\beta$ through the distance matrix $\mathbf{D}\^\mathrm{(F)}\_{i,j} = \||\alpha\_i\^\mathrm{(F)} - \beta\_j\^\mathrm{(F)}\||\_1=\||(\alpha_i - \beta_j)\mathbf{W}\^\mathrm{(F)}\||\_1$. The KL terms does not depends on $\alpha$ and $\beta$, since the optimum $\mathbf{T}$ and the $\mathbf{1}\_m$, $\mathbf{1}\_n$, ${\Delta}\_m$, ${\Delta}\_n$ are constants. Using the chain rule, we have:
> $$\frac{\partial \mathcal{P}}{\partial \alpha\_i} = \sum\_{j=1}\^\mathrm{m} \frac{\partial \mathcal{P}}{\partial \mathbf{D}\^\mathrm{(F)}\_{i,j}} \frac{\partial \mathbf{D}\^\mathrm{(F)}\_{i,j}}{\partial \alpha_i} = \sum\_{j=1}\^\mathrm{m} \mathbf{T}\_{i,j} \frac{\partial \mathbf{D}\^\mathrm{(F)}\_{i,j}}{\partial \alpha\_i} = \sum\_{j=1}\^\mathrm{m} \mathbf{T}\_{i,j} \mathbf{W}\^\mathrm{(F)} \cdot \mathrm{sign}(\mathbf{e}\_{i,j})\^\top,$$
> $$\frac{\partial \mathcal{P}}{\partial \beta\_j} = \sum\_{i=1}\^\mathrm{n} \frac{\partial \mathcal{P}}{\partial \mathbf{D}\^\mathrm{(F)}\_{i,j}} \frac{\partial \mathbf{D}\^{(F)}\_{i,j}}{\partial \beta\_j} = \sum\_{i=1}\^\mathrm{n} \mathbf{T}\_{i,j} \frac{\partial \mathbf{D}\^\mathrm{(F)}\_{i,j}}{\partial \beta\_j} = -\sum\_{i=1}\^\mathrm{n} \mathbf{T}\_{i,j} \mathbf{W}\^\mathrm{(F)} \cdot \mathrm{sign}(\mathbf{e}\_{i,j})\^\top.$$
> where $\mathbf{e}\_{i,j}=(\alpha\_i - \beta\_j)\mathbf{W}\^\mathrm{(F)}\in\mathbb{R}\^\mathrm{T}$, $\mathrm{T}$ is the sequence length, $\mathbf{e}\_{i,j,k}$ denotes the $k$-th element of $\mathbf{e}\_{i,j}$, the derivation of $\frac{\partial \mathbf{D}\^\mathrm{(F)}\_{i,j}}{\partial \alpha\_i}$ and $\frac{\partial \mathbf{D}\^\mathrm{(F)}\_{i,j}}{\partial \beta\_j}$ are detailed in our revised appendix.
>   - **Action.** **We have added `Theorem C.6` to formulate the calculation of gradients.** The gradient formulations are also refined to accommodate the 1-norm formulation.

---

> > ### Comment · Reviewer_4Wtk · 2024-11-22
> >
> > Thank you for you detailed reply to my comment.
> >
> > > The KL terms does not depends on \alpha  and \beta, since the optimum T ...
> >
> > Why does this hold? Since the optimum T depends on alpha and beta. I guess dT/dalpha and dT/dbeta should be considered.
> >
> > 4Wtk

---

> ### Author Response · Authors · 2024-11-20
>
> #### **[Q5] After (1), D is squared distance, while in definition 3.1 D is distance without square. Which is correct?**
> **Response.** We sincerely apologize for this typo and appreciate the chance to correct it. Given $\alpha_i$ and $\beta_j$, the correct definition of $\mathbf{D}$, as stated in Definition 3.1, is the standard (non-squared) Euclidean distance: $\mathbf{D}_{i,j} = \||\alpha_i - \beta_j\||$. Additionally, $\mathbf{D}\_{i,j}^{\mathrm{(F)}} = \||\alpha_i^{\mathrm{(F)}} - \beta_j^{\mathrm{(F)}}\||_1$, where the absolute error enhances robustness against large deviations, accommodating the characteristics of spectral data with energy concentrated in specific frequency bins.
> - **Action.** We have revised the manuscript to rectify the definition of $\mathbf{D}$ and $\mathbf{D}^\mathrm{(F)}$.
>
> #### **[Minors] In the end of p3: Fig. 4 should be Fig 1? In Fig 1(a), the left is W^(F)? The first word of Sec3.2: 'time-series' -> 'Time-series'**
> **Response.**  We appreciate the reviewer’s attention to these details. We have corrected the reference from Fig. 4 to Fig. 1 at the end of page 3. In Fig. 1(a), we have clarified that the left component represents $\mathcal{W}^{(F)}$ by updating the figure legend accordingly. We have capitalized “Time-series” at the beginning of Section 3.2.  Big thanks for your meticulous reading!
>
> ---
> **Reference**
>
> [1] Rethinking the Diffusion Models for Missing Data Imputation: A Gradient Flow Perspective, NeurIPS 2024.

---

> ### Author Response · Authors · 2024-11-22
> **Further clarifications on gradient calculation**
>
> Thank you very much for reading our rebuttal and raising follow-up questions.
>
> > Since the optimum T depends on alpha and beta. I guess dT/dalpha and dT/dbeta should be considered.
>
> **Response.**  We understand that the derivatives $dT/d\alpha$ and $dT/d\beta$ ideally should be considered. However, T is obtained through an iterative numerical optimization procedure $\mathcal{A}$ that is not always differentiable. Even if $\mathcal{A}$ was differentiable, calculating $dT/d\alpha$ and $dT/d\beta$ would be computationally intensive due to the need to compute gradients across up to 1,000 iteration steps. Even worse, large values in iteration could cause excessively large gradients and lead to gradient explosion. Therefore, we choose to update $\alpha$ and $\beta$ through the pathway $\mathcal{P}\rightarrow \mathbf{D}\^\mathrm{(F)}\rightarrow \alpha,\beta$. This strategy stabilizes the optimization procedure and effectively guides the imputation updates. We have clarified this point in Theorem C.6.

---

> > ### Comment · Reviewer_4Wtk · 2024-11-22
> >
> > Thank you for your answer. I think it must be clarified in the main text that the equations below 'Backward Pass' are not a true gradient of (2). Because it causes a risk that readers (especially students) may misunderstand that, in general, gradient of the optimal value (P) can be derived without considering the dependence between optimal solution (T) and optimization problem parameters (alpha,beta).
> >
> > 4Wtk

---

> ### Author Response · Authors · 2024-11-22
> **The clarification on gradient calculation has been incorporated in the manuscript.**
>
> Thank you very much for your swift response. We have incorporated the clarification into the main text. Specifically, on page 5, footnote 1, we note:
> > The definition of the DFT matrix is presented in Definition C.5, and the gradient derivation is detailed in Theorem C.6. Notably, our gradient formulation omits the gradients from the optimal $T$ with respect to $\alpha$ and $\beta$ to enhance the efficiency and stability of the calculation process.

---

> ### Author Response · Authors · 2024-11-27
>
> Dear reviewer 4Wtk,
>
> Thank you very much for your valuable feedback and insightful suggestions. *Following your recommendation, we have incorporated clarifications on gradient calculation into the main text. The revised manuscript highlights these changes in blue for your convenience.*
>
> As the discussion draws to a close, we hope to know whether you have *any remaining questions or additional suggestions*.
>
> Please note that today is the final day to submit the revised PDF. If you have any further comments, please kindly consider communicating them to us, such that we could promptly discuss them with you and reflect them in the revision.
>
> Thank you in advance,
>
> 14099 Authors

---

> ### Author Response · Authors · 2024-11-30
>
> Dear Reviewer 4Wtk,
>
> We sincerely appreciate the time and effort you have dedicated to reviewing our manuscript. Your discussions and suggestions have been invaluable in enhancing our work.
>
> In response to your comments, we have made the following efforts:
> - **For Q1**: we have clarified that the masking matrix is solely used for splitting test data.
> - **For Q2**: we have justified noise robustness by comparing Lemma 3.2 and Theorem C.2 (highlighted in blue fonts in revision).
> - **For Q3**: we have explained the efficacy of KL divergence without normalization.
> - **For Q4**: we have elaborated on the DFT matrix and the calculation of gradients (highlighted in blue fonts in revision).
> - **For Q5**: we have corrected typographical errors (highlighted in blue fonts in revision).
> - **For W2**: we have refined the technical motivation narrative in the introduction (highlighted in blue fonts in revision).
>
> As the discussion deadline approaches, we are wondering whether our responses have properly addressed your concerns? Your feedback would be extremely helpful to us. If you have further comments or questions, we hope for the opportunity to respond to them.
>
>
> Many thanks,
>
> 14099 Authors

---

> ### Author Response · Authors · 2024-12-02
>
> Dear reviewer 4Wtk,
>
> Since the discussion period will end in around a day, we will be online awaiting your feedback on our discussion, which we believe has fully addressed your concerns.
>
> We would highly appreciate it if you could take into account our discussion and clarification when updating the rating and having discussions with AC and other reviewers.
>
> Thank you so much for your time and efforts. We understand that the recent Thanksgiving holiday may have caused our previous message to be unintentionally overlooked. Sincerely sorry for our repetitive messages, but we're eager to ensure everything is addressed.
>
>
> Authors of #14099

---

### Official Review · Reviewer_8uNw · 2024-11-04

**Soundness:** 3
**Presentation:** 2
**Contribution:** 3
**Rating:** 8
**Confidence:** 2

**Summary:**

The paper proposes a time series imputation method based on optimal transport. The key idea is the combination of a frequency-based Wassertein discrepancy  and selective matching regularization. Theoretical justification is also provided. The experimental results show the imputation accuracy outperforms many sota methods.

**Strengths:**

1. An interesting and well-motivated design of the spectral-enhanced Wasserstein distance (WD)
2. A theoretical justified design of proximal spectral WD to account for non-stationarity.
3. seemingly excellent performance in real-world benchmark datasets.

**Weaknesses:**

1. The biggest issue from my end is the lack of standard deviation. From Table 1, the error of the proposed method seems really good, but i am not informed if these results are averaged over multiple train/test runs or just one run. To avoid cherry picking, the authors are encouraged to highlight how these numbers were obtained, what the training/test splits were, and what hyperparameter selection/cross-validation process was involved, etc.  Similar expectations apply to table 3 and 4.

2. Lack of convergence discussion/analysis. From Fig. 3 and Section 3.4, the imputation procedure seems to repeatedly sample patches from the time series, compute PSW, and use the gradient of the PSW to update the imputation. There seems a lack of convergence guarantees or discussions about this procedure. Can authors provide at least some discussion on this?

3. Data noise issue. Real time series data often includes noises. That means, just computing the distance after DFT in lin154 might be affected by data noises. Have the authors consider any methods or trade-offs, such as low-pass filters, in your SWD definition, to improve the robustness and/or counteract the noise effect?

**Questions:**

see above.

---

> ### Author Response · Authors · 2024-11-20
>
> **Thank you very much for your positive comments and appreciation of our novelty, motivation and theoretical & empirical results. Below are our responses to the specific concerns and queries.**
>
> ---
>
> #### **[W1] The biggest issue from my end is the lack of standard deviation. From Table 1, the error of the proposed method seems really good, but i am not informed if these results are averaged over multiple train/test runs or just one run. To avoid cherry picking, the authors are encouraged to highlight how these numbers were obtained, what the training/test splits were, and what hyperparameter selection/cross-validation process was involved, etc. Similar expectations apply to table 3 and 4.**
> **Response.** We understand that the standard deviation and the detailed description of dataset split and hyperparameter selection is critical for reproduction. The detailed responses to each query are provided as follows.
> - **Firstly, we have conducted additional experiments to provide the standard deviation of Table 1, 3, and 4. Please kindly check them in the subsequent comment windows.** We agree that standard deviations are important to convince the readers of the reliability of experiments and conclusions. **Therefore, we have added them in the revised appendix.**
> - **Secondly, we split the dataset** using the binary mask matrix, denoted as $\mathbf{M}$ in Section 2.1. The binary mask matrix has the same shape as the dataset and is generated by sampling a Bernoulli random variable with a predefined mean (i.e., missing ratio). Each element in the binary mask matrix takes a value of 1 or 0, representing whether the corresponding index in the dataset is considered missing or observable, respectively.
>   - For indices where the mask matrix element is 1, these indices are treated as missing in the training data. The associated data are excluded from the model training process and preserved exclusively for testing.
>   - For indices where the mask matrix element is 0, these indices are treated as observable in the training data. The associated data are used for model training.
> - **Thirdly, we introduce the hyperparameter selection process.** We leave out 5\% indices from the training data for validation set. Subsequently, the key hyperparameters involved in PSW-I are tuned in the validation set by minimizing the MSE, where the patch size is tuned within $\{24, 36, 48\}$; the matching strength is tuned within \{1, 10, 100, 1000\}.
>
>
> #### **[W2] Lack of convergence discussion/analysis. From Fig. 3 and Section 3.4, the imputation procedure seems to repeatedly sample patches from the time series, compute PSW, and use the gradient of the PSW to update the imputation. There seems a lack of convergence guarantees or discussions about this procedure. Can authors provide at least some discussion on this?**
>
> - **Response.** Thank you for your query, and we agree that the convergence property deserves detailed discussion.
> - Firstly, the PSW, denoted as $\mathcal{P}^\kappa$, is a convex function w.r.t. the input pair $(\alpha,\beta)$. It immediately follows from the fact that the composition of convex functions is convex. The PSW is convex (linear function is convex) to the pair-wise distance $\mathbf{D}^\mathrm{(F)}$ and $\mathbf{D}^\mathrm{(F)}$ is convex (1-norm and affine functions are convex) to $\alpha$ and $\beta$. Therefore, $\mathcal{P}^\kappa$ is convex to the input pair.
> - **Thanks to the convexity of $\mathcal{P}^\kappa$, it is feasible to prove that PSW-I, which employs gradient descent to update $(\alpha,\beta)$, converges (in Theorem C.3) with diminishing errors (in Theorem C.4).** Specifically, suppose $(\alpha_k, \beta_k)$ is the imputation result at the k-th iteration, $(\alpha^*, \beta^*)$ are the optimum; in Theorem C.4, we demonstrate that
> $$\mathcal{P}(\alpha_{k}, \beta_{k})-\mathcal{P}(\alpha^*,\beta^*) \leq \frac{1}{2\eta\mathrm{k}}\epsilon^2$$
> where as $k$ increases, the error diminishes with rate $\mathcal{O}(1/k)$

---

> ### Author Response · Authors · 2024-11-20
>
> #### **[W3] Data noise issue. Real time series data often includes noises. That means, just computing the distance after DFT in lin154 might be affected by data noises. Have the authors consider any methods or trade-offs, such as low-pass filters, in your SWD definition, to improve the robustness and/or counteract the noise effect?**
>
> **Response.** This is indeed a very insightful comment for extending PSW-I, which actually coincides the original objective of this research. We are pleased to share our considerations and trade-offs in this regard.
> - **This extension is theoretically promising**. Noise predominantly occupies the high-frequency components of the signal, whereas the underlying data semantics are primarily contained within the low-frequency regions. By applying a low-pass filter before calculating the spectrum distance between patches, it is feasible to attenuate the noise while preserving the essential semantic information. Consequently, this approach facilitates accurate time series imputation even amidst noisy observations.
> - **This extension is empirically valid**. To verify the extension you suggested, we have added additional experiments. Specifically, we introduce Gaussian noise to the observed data in frequency components above 100 while keeping the test data clean. As shown in the table below, most models experience a performance decline. To counteract noise, PSW-I+ applies a low-pass filter with a cutoff frequency of 20 prior to computing PSW. Experimental results demonstrate that PSW-I+ consistently outperforms PSW across all cases. However, the improvement at this stage is modest, potentially because the benefits of noise reduction are offset by the loss of semantic information in the high frequencies. Further performance enhancements may entail careful tuning of the low-pass filter, particularly the cutoff frequency.
> - **Some trade-offs hinder us from incorporating this extension. Specifically, it is difficult to tune the filter's hyperparameters**, typically the stopping threshold. In practice, the specific frequency bands affected by noise is not always available. Additionally, the absence of clean data makes it challenging to use a validation set for tuning the filter's hyper-parameters. Due to these challenges to determine the filter's hyperparameters, we have not extended our method to accommodate noisy datasets in the current study. Addressing these challenges requires further research, potentially involving adaptive filtering techniques or leveraging unsupervised learning methods to estimate the noise characteristics.
>
> | Dataset | ETTh1 |  | ETTh2 |  | ETTm1 |  | ETTm2 |  | Illness |  |
> |---|---|---|---|---|---|---|---|---|---|---|
> | Model | MSE | MAE | MSE | MAE | MSE | MAE | MSE | MAE | MSE | MAE |
> | DLinear | 0.172 | 0.27 | 0.109 | 0.233 | 0.105 | 0.222 | 0.079 | 0.197 | 0.25 | 0.298 |
> | TimesNet | 0.23 | 0.338 | 0.121 | 0.251 | 0.064 | 0.175 | 0.07 | 0.188 | 0.299 | 0.322 |
> | FreTS | 0.197 | 0.319 | 0.108 | 0.227 | 0.051 | 0.152 | 0.039 | 0.135 | 0.306 | 0.36 |
> | PatchTST | 0.182 | 0.278 | 0.14 | 0.271 | 0.053 | 0.154 | 0.045 | 0.147 | 0.91 | 0.584 |
> | SCINet | 0.178 | 0.285 | 0.123 | 0.247 | 0.069 | 0.178 | 0.061 | 0.170 | 1.147 | 0.711 |
> | iTransformer | 0.182 | 0.275 | 0.100 | 0.211 | 0.053 | 0.151 | 0.036 | 0.128 | 0.429 | 0.355 |
> | SAITS | 0.217 | 0.304 | 0.194 | 0.259 | 0.057 | 0.154 | 0.047 | 0.135 | 0.264 | 0.273 |
> | CSDI | 0.222 | 0.322 | 0.154 | 0.243 | 0.052 | 0.135 | 0.117 | 0.123 | 0.335 | 0.419 |
> | Sinkhorn | 1.002 | 0.754 | 1.009 | 0.752 | 1.004 | 0.755 | 0.996 | 0.749 | 1.034 | 0.721 |
> | TDM | 1.018 | 0.753 | 1.007 | 0.75 | 1.003 | 0.755 | 0.995 | 0.748 | 1.018 | 0.711 |
> | PSW-I | 0.169 | 0.257 | 0.051 | 0.145 | 0.049 | 0.131 | 0.023 | 0.092 | 0.111 | 0.157 |
> | PSW-I+ | **0.168** | **0.247** | **0.047** | **0.144** | **0.048** | **0.129** | **0.022** | **0.089** | **0.11** | **0.153** |

---

> ### Author Response · Authors · 2024-11-20
> **Standard deviations in Table 1, 3, and 4.**
>
> **The standard deviation of Table 3.**
>
> |Dataset| Distances | MAE  | MSE  | MRE  |
> |----|-----------|-------|------|------|
> |Electricity| PSW-T     | 0.007 | 0.010 | 0.006 |
> || PSW-A | 0.008 | 0.003 | 0.009 |
> || PSW-P | 0.008 | 0.002 | 0.003 |
> || PSW | 0.004 | 0.004 | 0.007 |
> |ETTh1| PSW-T     | 0.005 | 0.004 | 0.002 |
> || PSW-A   | 0.001 | 0.006 | 0.007 |
> || PSW-P  | 0.008 | 0.008 | 0.006 |
> || PSW  | 0.005 | 0.007 | 0.006 |
>
> **The standard deviation of Table 4.**
>
> |Dataset| Distances | MAE  | MSE  | MRE  |
> |----|-----------|-------|------|------|
> |Electricity| OT   | 0.008 | 0.002 | 0.003 |
> || EMD  | 0.008 | 0.006 | 0.007 |
> || UOT  | 0.004 | 0.000 | 0.008 |
> || Ours | 0.004 | 0.007 | 0.004 |
> |ETTh1| OT  | 0.005 | 0.006 | 0.004 |
> || EMD   | 0.003 | 0.010 | 0.003 |
> || UOT  | 0.001 | 0.004 | 0.008 |
> || Ours | 0.007 | 0.006 | 0.005 |
>
> **The standard deviation of Table 1. We only report the results of 4 models here. Full results are presented in Table 9 in revision.**
>
> | Datasets  | ETTh1 || ETTh2 || ETTm1 || ETTm2 || Electricity || Traffic || Weather || Illness || Exchange || PEMS03 ||
> |----|---|----|---|--|---|---|---|---|---|---|--|--|--|--|--|---|---|--|---|--|
> | Missing ratios   | MSE   | MAE   | MSE   | MAE   | MSE   | MAE   | MSE   | MAE   | MSE   | MAE   | MSE   | MAE   | MSE   | MAE   | MSE   | MAE   | MSE   | MAE   | MSE   | MAE   |
> | **Transformer** |    |   |   |    |  |  |  |  |  |  |  |||||||||
> | $p_{miss}=0.1$       | 0.0   | 0.004 | 0.008 | 0.008 | 0.009 | 0.005 | 0.007 | 0.005 | 0.004 | 0.005 | 0.009  | 0.009 | 0.01   | 0.008 | 0.0 | 0.004 | 0.004 | 0.004 | 0.002 | 0.001|
> | $p_{miss}=0.3$ | 0.003 | 0.005 | 0.002 | 0.003 | 0.003 | 0.008 | 0.007 | 0.005 | 0.003 | 0.001 | 0.008 | 0.004 | 0.003 | 0.008 | 0.008 | 0.008 | 0.0 | 0.002 | 0.001 | 0.0 |
> | $p_{miss}=0.5$ | 0.007 | 0.005 | 0.007 | 0.008 | 0.009 | 0.005 | 0.008 | 0.008 | 0.006 | 0.003 | 0.000 | 0.009 | 0.009 | 0.003 | 0.005 | 0.007 | 0.009 | 0.003 | 0.0 | 0.003 |
> | $p_{miss}=0.7$ | 0.005 | 0.002 | 0.004 | 0.000 | 0.007 | 0.008 | 0.002 | 0.007 | 0.002 | 0.0   | 0.001 | 0.002 | 0.007 | 0.007 | 0.006 | 0.001 | 0.006 | 0.0 | 0.003 | 0.002 |
> | **DLinear** |       |       |       |       |       |       |       |       |       |       |       |       |       |       |       |       |       |       |       |
> | $p_{miss}=0.1$      | 0.005 | 0.007 | 0.002 | 0.008 | 0.001 | 0.003 | 0.000 | 0.007 | 0.009 | 0.007 | 0.001 | 0.008 | 0.006 | 0.001 | 0.002 | 0.003 | 0.001 | 0.008 | 0.004 | 0.002 |
> | $p_{miss}=0.3$ | 0.006 | 0.005 | 0.008 | 0.000 | 0.008 | 0.004 | 0.001 | 0.007 | 0.003 | 0.009 | 0.010 | 0.002 | 0.007 | 0.006 | 0.005 | 0.003 | 0.007 | 0.006 | 0.005 | 0.003 |
> | $p_{miss}=0.5$ | 0.008 | 0.003 | 0.004 | 0.007 | 0.010 | 0.001 | 0.005 | 0.002 | 0.004 | 0.003 | 0.004 | 0.004 | 0.009 | 0.008 | 0.001 | 0.005 | 0.010 | 0.003 | 0.008 | 0.002 |
> | $p_{miss}=0.7$ | 0.009 | 0.001 | 0.002 | 0.008 | 0.007 | 0.007 | 0.007 | 0.003 | 0.010 | 0.009 | 0.007 | 0.004 | 0.004 | 0.001 | 0.008 | 0.005 | 0.007 | 0.006 | 0.007 | 0.008 |
> | **TimesNet** |       |       |       |       |       |       |       |       |       |       |       |       |       |       |       |       |       |       |       |
> | $p_{miss}=0.1$       | 0.008 | 0.006 | 0.009 | 0.006 | 0.004 | 0.009 | 0.005 | 0.006 | 0.001 | 0.009 | 0.004 | 0.010 | 0.005 | 0.002 | 0.003 | 0.007 | 0.009 | 0.006 | 0.003 | 0.009 |
> | $p_{miss}=0.3$ | 0.004 | 0.001 | 0.007 | 0.010 | 0.005 | 0.007 | 0.004 | 0.001 | 0.007 | 0.006 | 0.006 | 0.002 | 0.010 | 0.005 | 0.009 | 0.007 | 0.002 | 0.001 | 0.004 | 0.002 |
> | $p_{miss}=0.5$ | 0.008 | 0.005 | 0.005 | 0.0 | 0.002 | 0.00 | 0.001 | 0.010 | 0.003 | 0.004 | 0.007 | 0.002 | 0.003 | 0.001 | 0.005 | 0.006 | 0.008 | 0.007 | 0.005 | 0.007 |
> | $p_{miss}=0.7$ | 0.007 | 0.010 | 0.002 | 0.005 | 0.005 | 0.002 | 0.002 | 0.005 | 0.002 | 0.006 | 0.002 | 0.001 | 0.004 | 0.002 | 0.010 | 0.003 | 0.009 | 0.004 | 0.001 | 0.004 |
> | **PSW-I** |  |  |   |  |  |  |  |  |  |  |  |       |       |       |       |       |       |       |       |
> | $p_{miss}=0.1$       | 0.008 | 0.004 | 0.007 | 0.002 | 0.001 | 0.002 | 0.003 | 0.003 | 0.008 | 0.005 | 0.007 | 0.005 | 0.005 | 0.010 | 0.005 | 0.005 | 0.0 | 0.002 | 0.008 | 0.009 |
> | $p_{miss}=0.3$ | 0.004 | 0.008 | 0.005 | 0.002 | 0.009 | 0.009 | 0.004 | 0.009 | 0.002 | 0.001 | 0.002 | 0.004 | 0.004 | 0.003 | 0.007 | 0.004 | 0.001 | 0.005 | 0.009 | 0.000 |
> | $p_{miss}=0.5$ | 0.003 | 0.009 | 0.006 | 0.003 | 0.0 | 0.008 | 0.008 | 0.007 | 0.003 | 0.002 | 0.002 | 0.003 | 0.0 | 0.009 | 0.006 | 0.007 | 0.005 | 0.005 | 0.003 | 0.006 |
> | $p_{miss}=0.7$ | 0.004 | 0.006 | 0.001 | 0.003 | 0.003 | 0.009 | 0.006 | 0.002 | 0.002 | 0.009 | 0.002 | 0.005 | 0.002 | 0.001 | 0.007 | 0.006 | 0.009 | 0.004 | 0.009 | 0.007 |
>
> ---
>
> **20241122 Update:** We are pleased to announce that our implementation is now available here: https://anonymous.4open.science/r/psw-i-F35F. We hope this release reinforces your confidence and strengthens your support in this work.

---

> > ### Comment · Reviewer_8uNw · 2024-12-02
> >
> > Thanks for the new results and fruitful discussions. I've increased my score. I hope all these will be integrated into your paper.

---

> > > ### Author Response · Authors · 2024-12-03
> > >
> > > Thank you so much for reading our response and providing very positive feedback! These contents are incorporated in the revised manuscript. We commit preserving them which are fruitful to enhance the quality of this paper.

---

### Official Review · Reviewer_fN5T · 2024-11-04

**Soundness:** 3
**Presentation:** 4
**Contribution:** 3
**Rating:** 8
**Confidence:** 2

**Summary:**

Providing methods for time series imputations while respecting the temporal dependence.

**Strengths:**

- Very good empirical investigation.

- Clear presentation.

**Weaknesses:**

NA

**Questions:**

- What if the time series components are observed at different temporal frequencies (say days vs. hours, or hours vs. mins)?

---

> ### Author Response · Authors · 2024-11-20
>
> **Thank you very much for your encouraging support and appreciation of our presentation and empirical results. Below are our responses to the specific query raised.**
>
> ---
>
> #### **[Q1] What if the time series components are observed (a.k.a. sampled) at different temporal frequencies (say days vs. hours, or hours vs. mins)?**
> **Response.** Thank you for your insightful question. If you are interested in the general applicability of PSW-I to `different datasets with different sampling frequencies`, please refer to the results over the ETTh1 and ETTm1 datasets reported in our paper, which are observed at hourly and minute-level frequencies, respectively. If you hope to generalize PSW-I to `non-uniform time series`, which contain observations at mixed frequencies, we propose the following two solutions:
>
> - **Pre-Processing and Imputation.** A straightforward approach involves converting the non-uniform time series into a uniform time series before imputation. The process depends on the desired target frequency. For example, given a non-uniform time series sampled at both daily and hourly intervals: To recover an hour-wise sequence, the daily observations can be redistributed into hourly slots, with the unsampled hours treated as missing; to recover a day-wise sequence, the hourly observations can be aggregated or downsampled, and redundant indices can be removed. Once the time series has been resampled to the desired uniform frequency, standard imputation methods can be applied to handle any missing values.
>
> - **Using Non-uniform Discrete Fourier Transform (NDFT) (NDFT).** An alternative solution involves leveraging the NDFT to acquire the spectrum of non-uniform time series. By enhancing the pair-wise spectrum distance with NDFT, we can calculate the distance between non-uniform time series by comparing their spectrum, thereby extending the PSW-I framework to handle such data. The NDFT is defined as: $X_k = \sum_{n}^{N-1} x_n e^{-2\pi i p_n f_k}$, where $k$ (ranging from $0$ to  $N-1$ ) denotes the frequency index, $p_0, \dots, p_{N-1}$  are the scaled time stamps of the samples, and  $f_0, \dots, f_{N-1}$  are the corresponding frequencies.
>
> ---
>
> **20241122 Update:** We are pleased to announce that our implementation is now available here: https://anonymous.4open.science/r/psw-i-F35F. We hope this release reinforces your confidence in our work and inspires you to continue supporting our efforts.

---

> > ### Comment · Reviewer_fN5T · 2024-12-02
> >
> > Thank you for providing details on the noon-uniform time series case. I think this is a good piece of work and I will keep my score.

---

> ### Author Response · Authors · 2024-12-02
>
> Thank you so much for the continuing support -- we truly appreciate it!

---

### Author Response · Authors · 2024-11-27
**Summary by Authors**

We are grateful to all reviewers for their helpful comments regarding our paper. We are pleased that Reviewers 8uNw and 4Wtk appreciate the significance of methodology to enhance time-series imputation, finding our method novel and well-motivated. All reviewers find our empirical results promising. Below is a summary of our responses:

- To Reviewer fN5T: We have discussed a case where the time-series are irregularly sampled, and proposed some feasible extensions of PSW-I to accommodate it.
- To Reviewer 8uNw: We have included experiments across different random seeds, clarified the hyper-parameter selection process, underscored the convergence analysis, and conducted additional experiments with noisy dataset.
- To Reviewer 4Wtk: **After thorough discussion, the only remaining concern was the need to add a clarification on gradient calculation in the main text. This is an actionable item, and we have addressed it by including the necessary clarification on page 5, footnote 1.** Additionally, we have thoroughly refined the technical justifications and the narrative of the introduction to better highlight the fundamental technical concepts. **Therefore, we believe that all concerns raised by Reviewer 4Wtk have now been satisfactorily resolved.**

Please refer to our detailed responses to each point raised. We have also uploaded a revised manuscript with changes highlighted in blue. Hopefully our revisions and clarifications reinforce your confidence and support in our work.

Thank you again for your valuable time and expertise.

---

### Public Comment · ~Prabhant_Singh2 · 2025-04-07
**Code request**

Dear authors,

Thanks for a great paper; it was a really good read. Is it possible to get the code? I noticed that the code on github is deleted, Is there a new repository link?
Thank you

---

> ### Public Comment · ~Hao_Wang28 · 2025-04-07
>
> Thank you very much for your interest and appreciation.  Some refinement is needed to enhance the quality and exposure of the repo before official releasing. We will open a thread to perform it ASAP after completing an approaching deadline.

---

### Meta-Review · Area_Chair_nFmG · 2024-12-22

**Metareview:**

This submission tackles missing data imputation for time series via a new discrepancy measure. They target some difficult issues such as non-stationarity and periodicities, showing that the proposed proximal spectrum Wasserstein allows flexible learning and is amenable to the imputation framework. Reviewers concerns seemed addressed after the rebuttal phase. In particular, they already found the work novel with theoretical support and some questions and technical details were largely resolved.

**Additional Comments On Reviewer Discussion:**

Adequate discussion in response period

---

### Decision · Program_Chairs · 2025-01-22

Accept (Poster)